# Market-oriented job skill valuation with cooperative composition neural network

Ying Sun[1,2,3], Fuzhen Zhuang [1,4 ✉], Hengshu Zhu [2 ✉], Qi Zhang [2,5], Qing He [1,3] & Hui Xiong[6 ✉]

The value assessment of job skills is important for companies to select and retain the right talent. However, there are few quantitative ways available for this assessment. Therefore, we propose a data-driven solution to assess skill value from a market-oriented perspective. Specifically, we formulate the task of job skill value assessment as a Salary-Skill Value Composition Problem, where each job position is regarded as the composition of a set of required skills attached with the contextual information of jobs, and the job salary is assumed to be jointly influenced by the context-aware value of these skills. Then, we propose an enhanced neural network with cooperative structure, namely Salary-Skill Composition Network (SSCN), to separate the job skills and measure their value based on the massive job postings. Experiments show that SSCN can not only assign meaningful value to job skills, but also outperforms benchmark models for job salary prediction.

[1] Key Lab of Intelligent Information Processing of Chinese Academy of Sciences (CAS), Institute of Computing Technology, CAS, Beijing, China. [2] Baidu Talent Intelligence Center, Baidu Inc., Beijing, China. [3] University of Chinese Academy of Sciences, Beijing, China. [4] Institute of Artificial Intelligence, Beihang University, Beijing, China. [5] School of Computer Science, University of Science and Technology of China, Hefei, China. [6] Rutgers, the State University of New Jersey, Newark, NJ, USA. ✉email: zhuangfuzhen@ict.ac.cn; zhuhengshu@baidu.com; hxiong@rutgers.edu

In the era of knowledge economy, skilled talents are always precious treasures. Modern jobs require talents to have substantial and continuous investment on their job skills[1–3]. Therefore, understanding the value of job skill will fulfill the so-called "Skill Gap"[4,5] between employers and talents, and bring them competitive edge to cope with the accelerating pace of technological changes. At the micro level, it can not only help individuals to proactively assess their competencies and decide what are the right skills to learn, but also help companies to develop the right salary system of their job positions for attracting and retaining the best possible talent. Moreover, at the macro level, the job skill value is an important indicator of the economic equilibrium of labour market and shows the supply and demand relationship associated with knowledge investments[6].

During the past decades, researchers have devoted large efforts to assess the value of job skills in different manners. Many surveys and studies have shown evidence of a worldwide positive association between the distributions of job skill mastery and job salary[2,3,7,8]. However, due to the dynamic and indistinct nature of job skill value, traditional market survey-based approaches usually fail to provide a fine-grained and up-to-date analysis. In recent years, the newly available online recruitment services have accumulated abundant job advertisement data[9,10], which provides an unparalleled chance for Labour Market Intelligence[11,12] and data-driven job skill analysis[13,14]. Nevertheless, most existing studies are focused on job skill demand modeling[4,5,15,16], but there still lacks a quantitative way to assess the value of job skills from the perspective of their influence on job salary.

Indeed, achieving quantitative job skill value assessment is far from a trivial task. Specifically, on one hand, the value of a specific skill is not immutable but varies with respect to different job contexts. For example, the talents experienced with algorithm related skills will be appreciated with high-paid jobs for a high-tech AI company, while the engineering skills may be the most valuable ones in a traditional software company. On the other hand, the job skills are usually not isolated, but integrated with each other as a holistic requirement for deciding the job salary. Indeed, along this line, the most critical challenge is that there usually lack of ground truth data of skill value for building an effective and quantitative assessment model. Therefore, how to separately assess the value of job skills and model their impact on job salary under various job contexts is still open to be explored.

To this end, in this paper, we propose a data-driven solution to skill value assessment from a market-oriented perspective through mining the job advertisement data. Specifically, we introduce a market-oriented definition of skill value, and formulate the task of skill value assessment as the Salary-Skill Value Composition Problem, where each job position is regarded as the composition of a set of required skills attached with the job's contextual information, and the job salary is assumed to be influenced by the context-aware value of these skills. Along this line, we propose an enhanced neural network with cooperative structure, namely Salary-Skill Composition Network (SSCN), to separate the job skills and measure their value from the massive job postings. SSCN regards salary prediction as a cooperative task for skill valuation and holistically models the relationship between skills and the job salary, considering both skill value and domination. Figure 1 shows the schematic diagram of the key idea in this study. Indeed, SSCN provides a cooperative framework to train neural network models for knowledge discovery from unlabeled data, by quantitatively linking them with a supervised learning task. Extensive experiments on a real-world dataset clearly validate that SSCN can not only assign meaningful value to job skills in various job contexts, but also outperforms state-of-the-art models in terms of job salary prediction.

Meanwhile, based on the results of SSCN, many interesting findings can be revealed, such as which skills will lead to high-paid jobs.

As a long-standing research direction, the value of job skills in the market is always abstract and has different measurements with respect to different application scenarios[4,17]. Different from existing studies, in this paper, we aim to introduce a market-oriented definition of skill value with job context awareness, emphasizing the direct impact of skills on job salary. To be specific, the value of a skill is defined as the expected salary of a job that only requires this skill, given a specific job context. It should be noticed that in this paper, context refers to all the factors other than the skill requirement that can influence the job salary, such as the company, recruitment time, work location, and required working experience.

Indeed, the above definition directly measures how much salary a skill will bring when people make full use of it in the job. The motivation behind this definition is to guarantee that the value of different skills can be measured in an independent and comparable manner. In order to precisely estimate this value under various job contexts, we train a model $f$ with parameter $\Theta$ that calculates the skill value $v = f(s, \mathrm{lv}, \mathbf{C}|\Theta)$ given a set of observable job contexts $\mathbf{C}$ and a skill $s$ with level lv (i.e., the degree of mastery, refer to Fig. 2a for examples). To train the model, it is essential to obtain a set of training data containing job postings that only require one skill. However, in the real-world scenario, the job requirements are always complicated and cannot be qualified with only one skill. As a result, each job posting is always associated with multiple required skills, which makes it difficult to train the skill valuation model under the supervised learning paradigm.

Fortunately, the job salary can be regarded as a mixed value of corresponding required skills, and a job requiring many valuable skills should have a high salary. This intuition implies effective supervision for skill value assessment in an indirect way. In other words, if we can model the relationship between skill value and job salary, we can use job salary data to supervise the training of skill valuation model. Specifically, the job postings can be formulated as $\mathcal{J} = \{(\mathbf{C_j}, \mathbf{S_j}, \mathbf{Y_j})|j = 1, 2, \cdots \}$, where $\mathbf{C_j}$ denotes a set of job contexts, $\mathbf{S_j}$ denotes required skill set, $\mathbf{Y_j}$ denotes the job salary. In particular, $\mathbf{S_j}$ consists of the corresponding skill-level pairs $\mathbf{S_j} = \{(s_j^{(i)}, \mathrm{lv}_j^{(i)})|i = 1, 2, \cdots \}$, where $s_j^{(i)}$ is a skill and $\mathrm{lv}_j^{(i)}$ is the level. If we have a model that can precisely estimate the salary $\mathbf{Y_j}$ of a job posting given the value of its required skills, a proper estimation on skill value can lead to a good estimation on the job salary. So in this paper, we regard job salary prediction as a cooperative task for skill valuation. Formally, we define the task of this paper as a Salary-Skill Value Composition Problem, which aims to jointly learn a context-aware skill value assessment model $f$: (skill, context $\rightarrow$ value) and a skill-based salary prediction model $g$: (<skill, value> $\rightarrow$ salary) from the job postings set $\mathcal{J}$. It should be noticed that, although there might exist more complicated relationships among job skills, context and salary, in the problem formulation, we only consider the skill value is context-aware and can be combined together in a linear way to reflect the job salary. In this way, our model can facilitate the measurement of the influence of contexts on individual skills as well as the influence of skills on job salary.

Based on the above, the salary of a job $j$ can be formulated as $\widetilde{y}_j = g(\{(s_j^{(i)}, \mathrm{lv}_j^{(i)}, v_j^{(i)})|i = 1, 2, \cdots \}, \mathbf{C_j}|\Phi)$, where $\Phi$ and $\Theta$ denote the parameters, $v_j^{(i)} = f(s_j^{(i)}, \mathrm{lv}_j^{(i)}, \mathbf{C_j}|\Theta)$. By comparing the predicted job salary with the real salary, both the skill value assessment model $f$ and skill-based salary prediction model $g$ can be trained simultaneously.

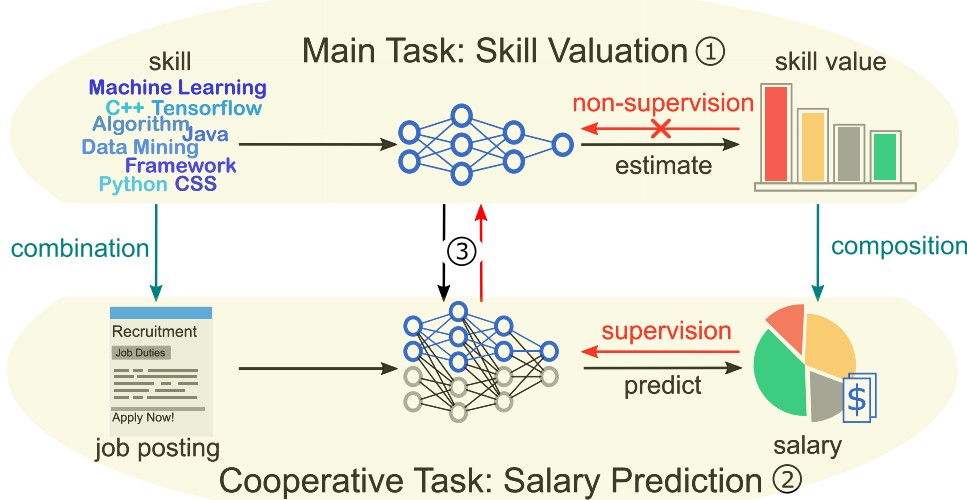

**Fig. 1 A schematic diagram of the key idea in this study.** (1) Our main task is to train a skill valuation model with machine learning technology. Under the paradigm of supervised learning, we need a set of training data with explicit labels of skill value to provide supervision for the model. Then the model can learn a function that maps the input (i.e., context and skills) to the observation (i.e., skill value). However, the labeled data of skill value is unavailable in our dataset. (2) We have abundant data of job postings with labels of salary, which can provide supervision for training a salary prediction model. Therefore, with the intuition that valuable skills should lead to high job salary, we regard salary prediction as a cooperative task that provides indirect supervision for skill valuation model. (3) We propose a model, SSCN, to simultaneously achieve skill valuation and salary prediction tasks, where the skill valuation model is a component of the salary prediction model. Specifically, SSCN estimates the skill value and composes skill value into job salary. In this way, the skill valuation model can be trained with feedbacks from the salary prediction task.

To solve the Salary-Skill Value Composition Problem, we propose the SSCN that is a cooperative neural network containing two steps of modeling to achieve skill valuation (the main task) and salary prediction (the cooperative task) simultaneously. The structure of SSCN is shown in Fig. 2b. Specifically, SSCN takes a job posting as the input, calculates the value of all the involved skills and then combines them into the job salary in a straightforward but interpretable way.

The first part of SSCN is a specially designed Context-aware Skill Valuation Network (CSVN), as shown in Fig. 2c. It dynamically models the skills, extracts the context-skill interaction and estimates the context-aware skill value. According to our definition, skill value can be regarded as a special case of job salary, and since salary is given as a range in our data, CSVN models the skill value as a range. Specifically, CSVN assigns each skill with a non-negative lower bound and a non-negative upper bound, constraining that the upper bound is no less than the lower bound.

In the real-world working scenario, the employees allocate their time and effort among the skills according to the importance of different job duties. Intuitively, the more you use a specific skill during work, the more it will influence your salary. Simulating this process, we propose to model the job salary as the weighted average of the skill value. We call the weight as skill domination. This agrees with our definition of skill value because when a job only involves one skill, the only skill has full domination and the salary degenerates into its value. In this way, the skill value is comparable and independent with each other. Considering that skills may have combinatorial influences on salary, we let the model catch skill interactions through modeling the domination. Specifically, the skill co-appearance is considered to influence the domination of each skill, which assures the model to peel explainable skill value that is only context-dependent while maintaining the model's fitting ability to general job postings. To model the domination, the second part of SSCN is a specially designed Attentive Skill Domination Network (ASDN), as shown

in Fig. 2d. Considering that the skill domination can be affected by the related skills (e.g., one skill may play an important role in the job when many related skills are also required), ASDN models the domination with a graph-based approach. Specifically, we attach each job posting with a skill graph, where the node represents the involved skills, and the edge between two skills represents their relationship. ASDN combines this skill graph with context-skill interaction information extracted from CSVN and calculates skill domination with graph-based attention mechanism. Considering that the two salary bounds may correspond to different job duty allocation, for example, common skills may raise the salary lower bound instead of the upper bound, ASDN outputs different skill domination for the two bounds. The details of training both CSVN and ASDN can be found in "Methods".

Indeed, SSCN models the relationship among skills, context and salary based on the observations of job advertisement data in an end-to-end manner. As a common issue of deep learning models, all the influencing factors and their complicated relationships are implicitly modeled as a blackbox, which is hard to be interpreted in a theoretical way. Nevertheless, it also brings the advantage that we only need to pay attention on the input (i.e., context and job skills) and output (i.e., job salary and skill value), while other latent influencing factors and relationships will be automatically learned by the hidden layers. In this way, the model is easy to be operated, and meanwhile, the skill value influenced by observable contexts can be explicitly estimated, which strongly supports further explainable analysis.

## Results

To validate the models proposed in this paper, we collected IT-related job postings from a popular online recruitment website in China, namely Lagou (https://www.lagou.com/). Our dataset contains over 800,000 postings of various job positions across a time span of 36 months, ranging from July 2016 to June 2019.

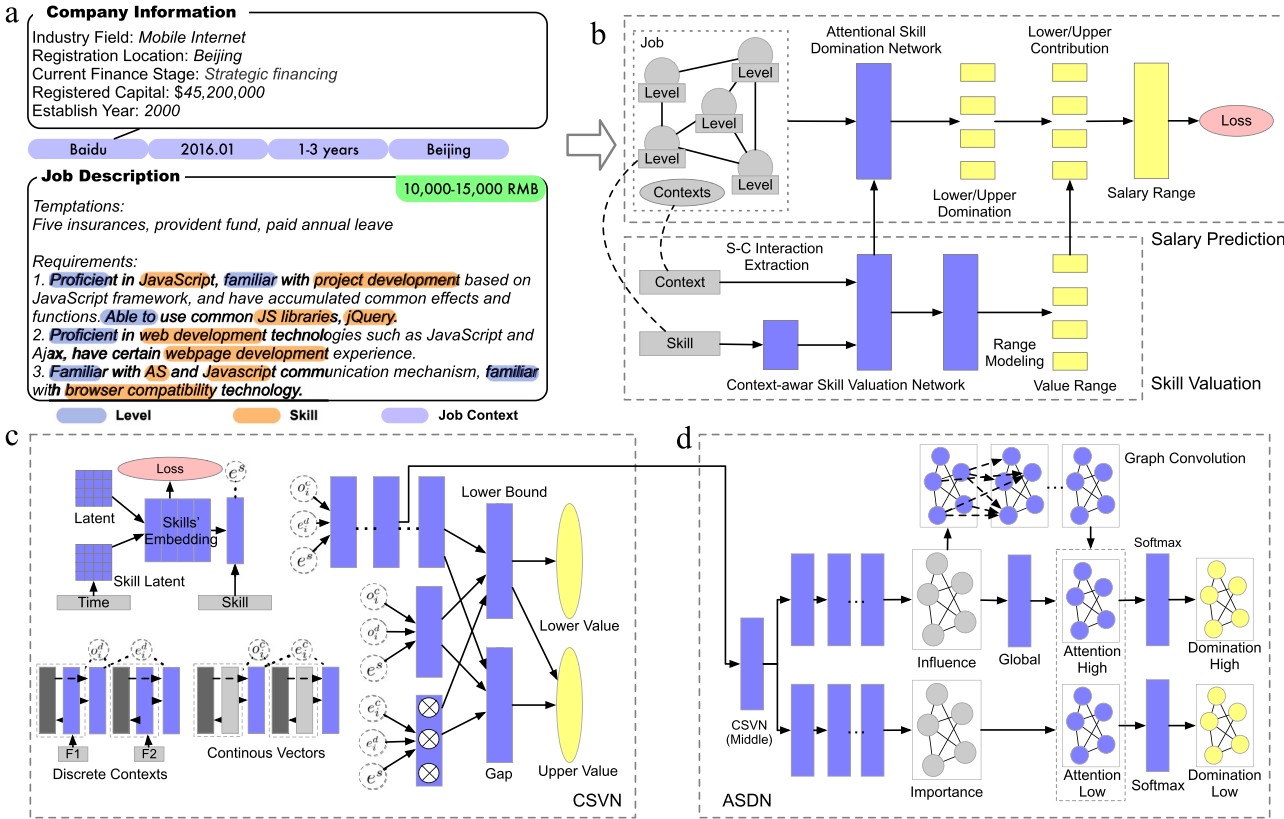

**Fig. 2 A schematic diagram of SSCN based skill valuation framework. a** An example of job posting in our data, which consists of some structured contextual information (e.g., company name, timestamp of publishing, work location, and required working experience), expected range of monthly salary (i.e., lower/upper bound in RMB), and detailed job description that introduces the requirements on candidates' job skills. In particular, each skill usually has a descriptive requirement on the degree of mastery, such as *Proficient* in *JavaScript*, and *Familiar* with *AS*. **b** We formulate the job posting as a set of skills formed in a skill graph, and some contextual inputs. Our proposed SSCN estimates skill value and combines them into the job salary. The color gray, blue, yellow and pink indicate inputs, model structures, outputs, and loss functions, respectively. **c** The detailed structure of CSVN. **d** The detailed structure of ASDN.

After filtering the data with some preprocessing steps, we got 215,308 samples. We used these samples to train and validate our model. The details of data preprocessing, feature selection, network configurations, numerical statistics, and additional experimental results can be found in Methods and Supplementary Information. In particular, we also conducted supplementary experiments on an additional designer-related job posting dataset to validate the generalization of our model.

**Skill value analysis under different job contexts**. Here we demonstrate the value of skills estimated by CSVN considering different kinds of job contexts. During our experiments, we found that the lower bound and upper bound of skill value always have a similar trend, so we mainly introduce the results of the lower bound, unless noted otherwise.

We define level influence as the average ratio of value increase when a level is specified. Figure 3a shows the levels' average influence (see Supplementary Fig. S8a for influence distribution), where we have used all the skill-level pair instances involving each level for the estimation. The detailed information on sample size and influence distribution can be found in Supplementary Table S10. We can observe that CSVN can significantly distinguish the impact of different levels. In general, most levels have a similar influence on both bounds, and sophisticated levels raise skill value more. In particular, the level *Can Read*, i.e., the lowest degree of mastery in our dataset, will decrease the skill value by 10%, while the level *Versatile* can contribute about 10% increase to the value. To get more insights, we show level influence on some specific

skills in Table 1. In addition, we conducted significance test for better validating the results. It can be observed that, by ignoring the insignificant entries (i.e., $p$-value > 0.05), the table is generally consistent with the averaged influence. Nevertheless, the model also learns bias for some special cases. For example, while *Know* is a relatively low level of mastery, it has positive influence on skill value when describing *JavaScript*. The reason is that while *JavaScript* mostly appears in jobs that related to web development, the statement *Know JavaScript* usually acts as an additional requirement for some complicated and higher-paid jobs like architecture design. Therefore, the model overestimates the skill value due to the imbalanced data distribution. Indeed, this result is explainable from a market-oriented view. Specifically, the mastery level of a specific skill usually indicates the role that it plays in the job; and therefore, the skill value highly depends on the market pricing on the relevant jobs. However, as shown in Fig. 3 (a), the model will still work for the general cases. Furthermore, we calculated the ratio of skill-level observations that might cause the biased level influence estimations. The result shows that only very few samples (0.96% of the whole dataset) encounter this bias. The detailed calculation can be found in the Supplementary Information. A possible solution for alleviating this kind of bias is to enlarge the diversity of the recruitment market data, which is a valuable direction for our future studies. Supplementary Fig. S6a shows the level influence on the designer dataset. The result slightly differs from the result on the IT dataset, which further indicates that level influence varies with respect to occupations.

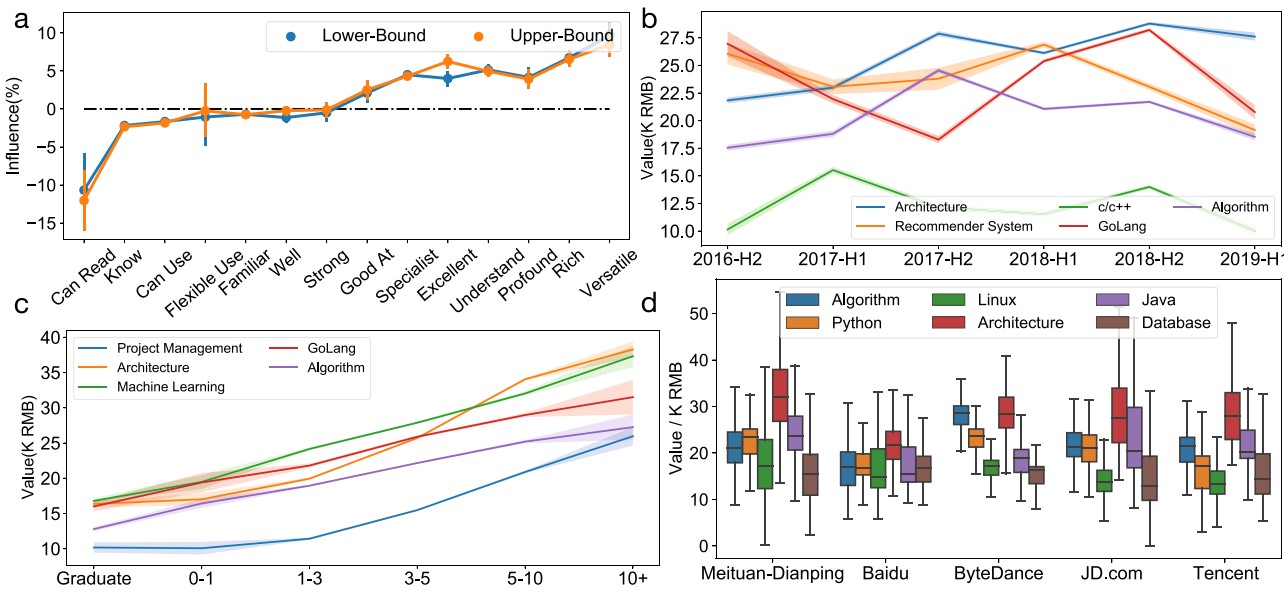

**Fig. 3 Skill valuation concerning different job contests. a** We calculated the influence of level lv as $r_s^{lv} = \frac{\sum_{i,j} \mathbb{1}\{lv_j^{(i)}=lv\}(v_j^{(i)}-v_{s_j^{(i)}})/v_{s_j^{(i)}}}{\sum_{i,j} \mathbb{1}\{lv_j^{(i)}=lv\}}$, where and $v_{s_j^{(i)}}$ denotes the averaged value of skill $s_j^{(i)}$. We also show the 95% confidence interval (CI) in the figure, where data are presented as mean values ± CI. We use different colors to indicate level influence on different bounds. **b** CSVN assigns the skills with temporal embeddings to catch their dynamic changes, we show the average value of some skills at different time intervals. The shadow shows the 95% confidence interval, where data are presented as mean values ± CI. We use different colors to indicate different skills. **c** The value of some randomly selected skills with different length of working experience. The shadow shows the 95% confidence interval, where data are presented as mean values ± CI. We use different colors to indicate different skills. **d** To analyze the value of skills with respect to different companies, we draw the value distribution of some popular skills in five famous Chinese Internet companies on boxplots. The box shows the quartiles of the dataset. The whiskers extend to show the rest of the distribution except for outliers. Specifically, as a common practice, we regarded the samples outside 1.5 times interquartile range (IQR) above the upper quartile or below the lower quartile as outliers. We use different colors to indicate different skills.

In this study, time is also regarded as one kind of job context. CSVN assigns the skills with temporal embeddings, this supports dynamic skill value analysis. From Fig. 3b, we can observe that fluctuations exist on skill value, and the skills have different trends of value change (see Supplementary Table S12 for numerical statistics). Some interesting findings can also be observed from the figure. On the whole, *Architecture* has a relatively stable trend of value increase. Specifically, in 2016-H2, its value is 21.8 K RMB on average. Then, it increased 5% on average for every half-year and reached 27.6 K RMB in 2019-H1. This indicates a rising market demand for this skill, which is good news for architects. However, some hot skills like *GoLang* and *Recommender System* seem to be less stable. Especially, *GoLang* has sharp value increase and decrease. For example, in 2019-H1, its value decreased by 26%, from 28.2 K RMB to 20.8 K RMB on average. This reminds students not to simply pursue the hottest new skills on the market, because their related industry may be still unstable. According to our experiment, we find that many skills with high value meet value decrease in the first half of 2019. We guess this phenomenon is due to the so-called Internet Winter of China this year. The trend of value for designer skills can be found in Supplementary Fig. S6b. Interestingly, the designer skills are stable and there is no general value decrease in the first half of 2019, which indicates that recent market changes have more influence on IT practitioners than designers.

Skill value under different experience requirements can provide talents with a long-term reference on choosing skills to learn. CSVN considers working experience requirements as one kind of job context and has a strong ability on inferring the experience-aware value, even for new skills. For example, although *GoLang* was officially released in 2009, we can still estimate its value with the working experience of longer than 10 years as 32.0 K RMB by smoothly extending the line. Figure 3c shows that longer experience leads to higher skill value (see Supplementary Table S13 for numerical statistics). Compared with the graduates, 10 years of working experience increases the skill value by 2.5 times on average. This is reasonable because a highly experienced talent usually can get a higher salary. But the speed of value rise has some differences among the skills. For example, *Architecture* and *Project Management* increase slowly in the first several years, while quickly after 3–5 years. Specifically, although *Algorithm* has a higher value (12.8 K RMB) for graduates, in the long term, the value of *Project Management* (10.2 K RMB for graduates) increases faster and achieves the similar value as *Algorithm* after 10 years. Similarly, *Machine Learning* has a higher value (16.8 K RMB) than *Architecture* (16.4 K RMB) for graduates and increases fast in the first several years. It can be observed that, with 1–3 years' experience, the value of *Machine Learning* (24.2 K RMB) is 20% higher than *Architecture* (19.9 K RMB). However, the rank is reversed after 5 years. This result makes sense, because the ability on *Architecture* and *Project Management* accumulates during work, while talents' programming skills usually gain fast the first several years of their career and may decrease as they get older. We can conclude that CSVN can provide good experience-aware skill value assessment. This provides students a reference to consider their longer future career when choosing a skill to learn, instead of only comparing the job salary at an entry-level. In addition to skills that get you a fortune at the moment you graduate from school, learning skills that are valuable for you in the future may also be a good choice. We also show the experience influence on designer skills in Supplementary Fig. S6c, which shows the similar trend with that of the IT dataset.

For job seekers, the best choice is to work in companies that treasure the skills they possess. Figure 3d shows skill value

**Table 1 The level influence on 6 kinds of programming skills.**

| Skill | Value | Know | p-value | Can use | p-value | Familiar | p-value | Understand | p-value | Rich | p-value |
|---|---|---|---|---|---|---|---|---|---|---|---|
| JavaScript | 15.74 | 8.11% | <0.001 | −0.33% | 0.835 | −1.27% | 0.034 | 3.88% | 0.365 | 8.61% | 0.326 |
| c/c++ | 12.58 | −9.33% | 0.026 | 1.93% | 0.413 | 0.15% | 0.856 | −14.67% | 0.210 | 9.29% | 0.040 |
| Python | 18.43 | −0.81% | 0.540 | −1.49% | 0.038 | −0.41% | 0.204 | −15.96% | 0.010 | 10.91% | 0.005 |
| scala | 21.41 | −0.57% | 0.913 | 15.73% | <0.001 | −3.19% | <0.001 | 13.42% | 0.609 | 10.59% | 0.485 |
| Java | 19.57 | −5.24% | <0.001 | −5.10% | <0.001 | −3.06% | <0.001 | 12.30% | <0.001 | 0.19% | 0.948 |
| c# | 9.35 | −10.86% | 0.127 | 10.73% | 0.023 | −1.50% | 0.411 | 61.15% | 0.051 | 65.44% | 0.006 |

We randomly picked some skills related to programming language and listed the influence of several levels on them. To better reveal the influence and distinguish occasional results, we conducted two-sided t-test for the significance of each skill-level pair in the table and listed the corresponding p-value.

distribution in different companies, where we have used all the skill-company pair instances involving each corresponding skill-company pair for the estimation. The detailed information on sample size and numerical statistics can be found in Supplementary Table S14. It can be observed that, due to the differences in business strategy, skills are valued differently by different companies. This reveals the traits of companies. For example, while most of these companies give a much higher value to *Architecture* than *Algorithm*, ByteDance values them similarly. Besides, ByteDance is the only company that values *Python* (23.9 K RMB on average) more than *Java* (21.0 K RMB on average). This implies ByteDance attaches high importance to some research works. In JD.com, *Java* has a larger range of value distribution than in other companies. Specifically, the gap between the two quartiles of *Java* in JD.com is 13 K RMB, which is much larger than the gaps of 7 K RMB in the other 4 companies. This implies the higher possibility of salary increase for a Java engineer in JD.com. Meanwhile, different from others, the value of skills in Baidu is quite stable, which means the demand for different skills is more comprehensive. In Supplementary Fig. S6d, we show the distribution of designer-related skill value on these companies. It can be observed that the companies also have different preferences in designer-related skills.

**Evaluation on salary prediction**. We compared the performance of SSCN on salary prediction with several baseline methods (see details in "Methods"). The performance is evaluated with root mean square error (RMSE) and mean absolute error (MAE)[18], which are both popular metrics for difference measurement between the observations and the predictions. The results of the evaluation are listed in Table 2. There are several observations. First, SSCN outperforms all the baseline models, especially in terms of RMSE where there is a 3.5% decrease on lower bound prediction and 5.2% decrease on upper bound prediction compared to BERT, which outperforms the rest of the baseline models. Though SSCN has a larger variance due to its complex structure, its worst performance is still significantly better than the others' best performances. Second, SSCN outperforms the linear models (i.e., SVM and LR). To assure the physical meanings of the skill value, SSCN simplifies the last layer of skill composition into a linear form. However, SSCN is still a complicated non-linear deep learning model that can seize the complicated relation between skill, context and salary. So it performs much better than the real linear models. Third, since accurately predict context-aware job salary is a more difficult problem than standard salary benchmarking, HSBMF performs not well. But SSCN can achieve more accurate salary prediction under specific job contexts. Fourth, by replacing ASDN with a mean pooling layer, the model's performance decreased a lot. This proves the effectiveness of skill domination on job salary modeling. Fifth, simultaneously estimating the two bounds of the range in a single model improves the performance. This is because the lower bound and upper bound of job salary are strongly correlated. In addition to giving constraints on the bounds, CSVN also extracts a shared shallow representation for them. In this way, the two bounds can get part of the supervision from each other, which reduces the chance of over-fitting. The experimental results on salary prediction on the designer dataset can be found in Supplementary Table S8, which are consistent with the results of the IT dataset. Furthermore, we conducted parameter experiments to demonstrate the robustness of our model, which can be found in Supplementary Fig. S5 and Supplementary Table S7. The results show that SSCN is parameter insensitive and can be easily adopted without carefully tuning the hyper-parameters.

**Table 2 Performance evaluation on salary prediction.**

| Model | Lower | | Upper | |
|---|---|---|---|---|
| | RMSE | MAE | RMSE | MAE |
| SVM | 5.675 ± 0.215 | 4.120 ± 0.028 | 10.404 ± 1.202 | 7.177 ± 0.038 |
| LR | 5.386 ± 0.021 | 4.033 ± 0.013 | 9.545 ± 0.049 | 7.139 ± 0.028 |
| GBDT | 4.878 ± 0.023 | 3.651 ± 0.017 | 8.763 ± 0.032 | 6.568 ± 0.027 |
| DNN | 6.498 ± 0.031 | 4.999 ± 0.036 | 11.801 ± 0.021 | 9.460 ± 0.020 |
| HSBMF | 5.291 ± 0.017 | 3.939 ± 0.015 | 9.188 ± 0.036 | 6.800 ± 0.028 |
| TextCNN | 4.999 ± 0.028 | 3.712 ± 0.018 | 8.800 ± 0.057 | 6.554 ± 0.057 |
| HAN | 4.761 ± 0.043 | 3.497 ± 0.054 | 8.333 ± 0.069 | 6.111 ± 0.092 |
| Transformer-XL | 5.459 ± 0.016 | 4.097 ± 0.045 | 9.663 ± 0.061 | 7.278 ± 0.074 |
| BERT | 4.592 ± 0.010 | 3.331 ± 0.011 | 8.110 ± 0.136 | 5.841 ± 0.137 |
| RoBERTa | 4.642 ± 0.014 | 3.377 ± 0.011 | 8.400 ± 0.076 | 6.122 ± 0.058 |
| XLNet | 4.566 ± 0.015 | 3.333 ± 0.011 | 8.254 ± 0.060 | 5.995 ± 0.044 |
| CSVN + Mean | 6.758 ± 0.041 | 5.085 ± 0.038 | 11.46 ± 0.118 | 8.640 ± 0.084 |
| SSCN (Independ) | 4.762 ± 0.063 | 3.484 ± 0.052 | 8.278 ± 0.091 | 6.021 ± 0.079 |
| SSCN | **4.435 ± 0.061** | **3.244 ± 0.048** | **7.686 ± 0.086** | **5.627 ± 0.060** |

Bold formatting indicates the best performance among all these models.
10 times of hold-out validation were repeated on each model, where we randomly split the data for training and testing with a ratio of 4:1 at each time. The results of RMSE and MAE are listed in the form of mean ± standard deviation.

It can be concluded that, with the cooperation of the salary prediction task, SSCN trains a quantitative and accurate skill valuation model without using any labeled skill value data. Since skill valuation is an essential component of job salary prediction in SSCN, SSCN's performance on job salary prediction also quantitatively demonstrated the effectiveness of our skill valuation model.

## Discussion

With the Salary-Skill composition structure, SSCN decouples the job salary into the value of every involved skills by modeling skill domination. Here, we analyze this composition process holistically and show the effect of its factors.

**Skill domination versus skill value**. The multiplication of value and domination of some skill in a job posting is its actual contribution to the salary. To analyze the effect of domination and value, we display the averaged value, domination, and salary contribution of machine learning-related skills in Fig. 4. The numerical statistics can be found in Supplementary Table S16–S18. On the whole, more generic skills have higher domination, while more specific skills have higher value. For example, *Unsupervised Learning* (with domination 37.8% on average) and *Multivariable Regression* (with domination 46% on average) have high domination, showing many jobs need them. *Graph Algorithm* (with domination 18.2% on average) has lower domination but higher value (with value 35.2 K RMB on average), indicating that although there are fewer jobs that can make full use of it, you can easily get high salary if you find one. Indeed, most jobs in the market are not so professional and are dominated by some generic skills. In these jobs, some high-value skills may also be involved, but they are usually not a major part of the work. Also, the rapidly-emerging new skills with the fast technology changes enlarge the skill gap between job candidates and employers[19]. As a result, from the viewpoint of the employers, although it is usually difficult to find candidates who perfectly meet their specific skill requirements, the talents owning generic skills are usually able to quickly learn and adapt to the required skills[20]. Accordingly, higher education in recent years have been focusing on teaching theoretical and basic knowledge, and cultivating students' learning ability and problem-solving skills rather than teaching specific skills[21]. This phenomenon enlarges the domination of more generic skills in the job market.

Our experimental result implies that the breadth of your knowledge decides how easy you can find a job, while the depth of your skill helps to raise your salary. In this way, it becomes a trade-off between domination and value when choosing a skill to learn, the averaged contribution becomes a good reference, as is shown in Fig. 4c, *Topic Model* (with contribution 8.5 K RMB on average) is a good learning choice. It should be noticed that having a low averaged domination does not mean the skill never dominates a job. When you have excellent knowledge of some specific skills (which is always true for Ph.D. students), you should be confident that you can find somewhere to make full use of your ability. Wordclouds for the designer dataset can be found in Supplementary Fig. S7, where we can distinguish generic and specific skills for designer-related jobs.

**The influence of skill on job salary**. For a skill required in a job posting, we can estimate its influence by calculating how much will the salary decrease if we remove this skill from the requirement. By fixing the domination of the other skills and getting their weighted average of value, the new salary can be estimated as $y' = \frac{y-v}{1-d}$, where $v$ and $d$ represents the value and domination of the removed skill. The ratio of decrease is $r = \frac{y-y'}{y}$, where $y$ denotes the previous job salary. In Table 3, we can observe that generally, high value and high domination lead to high influence. For example, *Matrix Calculation* has a high value and high domination, by dropping it, the job salary will decrease by 18.4% on average. According to this table, machine learning-related skills have positive influence on job salary. We will show in the next part that some skills may have negative influences on job salary.

**Case study on a job posting**. Everyone wants a job where they can give full play to their ability. However, the job descriptions may contain job duties both you are good at and not good at. Understanding the role of each required skill in a job can help job seekers to decide if a job is suitable for them. For each job posting, SSCN predicts the value of each skill under the specified context, calculates the skill domination based on the skill co-appearance, and finally combines the skill value into the job salary according

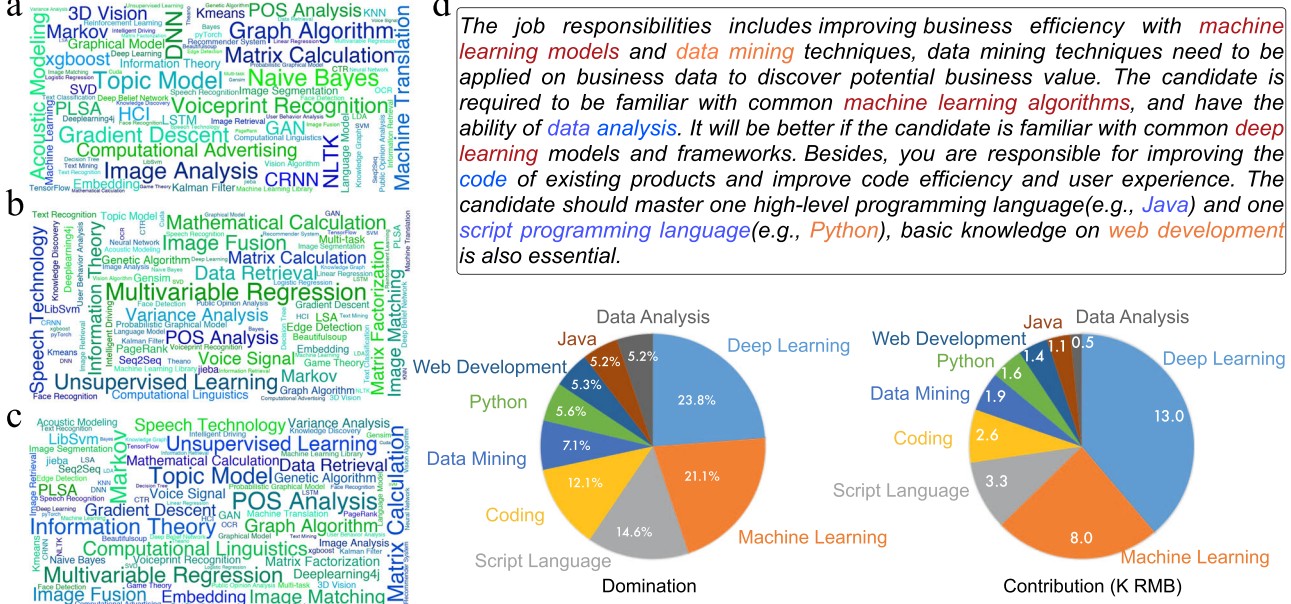

**Fig. 4 Visualizations of skill salary decompositions. a** We calculated the averaged context-aware skill value estimated by CSVN and drew word cloud of machine learning-related skills, where the size of each word shows the skill value. **b** Similar to the word cloud of skill value, we drew word cloud for averaged skill domination estimated by ASDN. **c** For each skill in a job posting, we can calculate the actual salary contribution of it by calculating the multiplication of its value and domination, and show the averaged contribution of each skill on the word cloud. **d** A case study on a job posting, the role of each skill is analyzed by calculating their domination, contribution, and influence on job salary. The color of words in the job description shows the skills' influence on salary, blue/yellow/red means the salary will increase/remain/decrease by dropping the skill. The pie plots show skill domination and contribution where the colors distinguish different skills.

| Table 3 Skill's average influence on salary. | | | | |
|---|---|---|---|---|
| **Skill** | **Salary** | **Value** | **Domination** | **Influence** |
| Matrix calculation | 19.872 | 32.306 | 25.2% | +18.4% |
| POS Analysis | 22.994 | 29.179 | 29.5% | +15.6% |
| Information theory | 21.425 | 26.794 | 28.7% | +13.7% |
| Computational linguistics | 20.725 | 24.632 | 23.8% | +12.0% |
| Voiceprint recognition | 20.694 | 33.235 | 10.2% | +8.8% |
| plsa | 23.600 | 28.043 | 14.5% | +4.7% |
| xgboost | 21.300 | 29.211 | 7.8% | +3.6% |

We show the averaged influence of some machine learning-related skills, together with their averaged value (K RMB) and domination, and the averaged salary (K RMB) of jobs involving them.

to the domination. Figure 4d shows the case study to illustrate how SSCN works on a job posting. Specifically, we used the trained SSCN to decompose a randomly selected job posting and analyzed the domination, contribution, and overall influence on salary of the involved skills. This job description is to employ an algorithm engineer who has two parts of job duties, which are data mining with business data and product development. Compared with the coding skills, *Machine Learning* and *Deep Learning* have much higher domination and contribution on the job salary, indicating that the job expects a data mining expert instead of an experienced engineer. Though with similar domination, *Deep Learning* has a much higher contribution than *Machine Learning*, which is because it has a higher value under its job context. We can also observe that *Deep Learning* contributes a lot to the higher-bound salary, which agrees with the job

description where *Deep Learning* is listed as the additional requirement. From the above analysis, we can find that the job seekers can try this job if they are good at machine learning and deep learning, there is no need to worry much if they are mediocre at coding. Also, it can be observed that data mining duty has a positive influence while the development duty has a negative influence on the salary. This indicates that if you are an expert in data mining, maybe you should look for a full-time data mining job, it may bring you a higher salary.

**Potential applications**. Through the experiments, we show that our skill valuation model has the potential to be applied to various real-world applications. First, our model can be applied to talent recruitment. As can be observed in Table 2, our model achieves high performance on salary prediction. Therefore, it can provide salary references for jobs in the market when the job descriptions are specified. With the predicted salary information, the recruiters can evaluate the market competitiveness of their offered salaries; and the job seekers can get an idea about their salary expectations. Second, our model can be directly applied to business market analysis. For example, as can be observed in Fig. 3b, our model reveals the overall trend of skill value in the market. Third, our model can be applied to student education. Specifically, the skill value provides the students with market-oriented guidance for skill learning. For example, with the experience-aware skill value shown in Fig. 3c, students are able to make better personalized curriculum choices to achieve long-term career development. Fourth, our model can be applied to knowledge management and talent development. For example, as shown in Fig. 3d, the companies can analyze the value of skills for their own business. Then, they can develop specific curriculums to continuously train their employees for valuable skills. Fifth, our model can be applied to job recommendation. For example, by

measuring the average value of skills in different companies, as shown in Fig. 3d, job seekers can receive effective guidance on which company is more suitable for them to pursue.

**Technical contribution**. Since indirect supervision is common in the real-world, we believe that this work not only provides an intelligent and accurate solution for the skill valuation problem but also can be an inspiration for readers who work on data analysis in other fields of applications. Specifically, in many real-world scenarios, obtaining labeled training data is far from an easy job. It is often the case that we can only obtain indirect supervision from a related task. Learning skill valuation model from job salary data is one of these kinds of problems. In this problem, we have no labeled data of skill value, but we have job salary data as indirect supervision information, with the intuition that high skill value usually leads to high job salary. To this end, we proposed a machine learning-based solution that uses neural network with cooperative structure to model the relationship between job and skills, where the salary prediction is regarded as a cooperative task for training the skill valuation model. In this way, we obtain an effective skill valuation model under the indirect supervision of job salary data.

**Limitations**. The first limitation of this paper is the limited data. On the one hand, since our work is based on the accumulated job advertisements in online recruitment website, which has a short history, we are not able to provide insights about the long-term job skill development. On the other hand, since our research has certain requirements on the data quality (e.g., detailed skill requirement, job salary and contextual information), in this paper we only evaluated our model with two datasets collected from one of the largest and most popular Chinese online recruitment website of Internet-related industry. This may bring bias to the analysis. If provided with more large-scale and comprehensive data, our model will obtain more significant insights. The second limitation is the empirical validation of skill value. Since market-oriented skill valuation is a new research problem, we are not able to obtain ground truth for quantitatively validating the accuracy of our model. Therefore, in this paper, we evaluated the performance of our model with the task of salary prediction. The rational behind our evaluation is that, with the explicitly formulated relationship between salary and skill, the effectiveness of skill value will be revealed from the salary prediction performance. In the future, we plan to continuously update our research by seeking more data sources and collaborations for further validating our model.

## Methods

**Job posting formulation**. A summary of the notations in this paper can be found in Supplementary Table S2. As shown in Fig. 2a, we formulate a job posting $\mathbf{J_j}$ as $(\mathbf{C_j}, \mathbf{S_j}, \mathbf{Y_j})$, where $\mathbf{C_j}$ denotes a set of job contexts, $\mathbf{S_j}$ denotes required skill set, and $\mathbf{Y_j}$ denotes the expected range of job salary. $\mathbf{S_j} = \{(s_j^{(i)}, lv_j^{(i)})|i = 1, 2, \cdots\}$ is a set of skill-level pairs involved in the job description, where $s_j^{(i)}$ is a skill and $lv_j^{(i)}$ is its level on the degree of mastery, for example *Proficient* in *JavaScript*. Considering that the relations between involved skills may affect job salary (e.g., a skill affects the job salary more if many skills related to it are also required), we attach $\mathbf{S_j}$ with a skill graph $\mathbf{A_j}$, where each node represents an involved skill and the edge weights represent the co-appearing relations between them.

**Data preprocessing**. We extracted 14 level words and 1374 IT-related skill words, so that the job descriptions can be formulated into structured records. The detailed descriptions of data preprocessing can be found in Supplementary Information. Then, we counted the co-appearing frequency of every two skills in the job advertisements. If the frequency is larger than a pre-defined threshold, we added an edge between these two skills, whose weight is the normalized co-appearing frequency. To reduce noise, we first filtered full-time job postings. Next, we ranked the cities according to the number of the samples they involve and filtered the job postings of the top 16 cities, which covers over 90% of the data. Then, we dropped

the records whose upper-bound or lower-bound salary is a boxplot outlier[22] in the dataset (see Supplementary Fig. S3 for the salary distribution). Finally, we ranked the companies according to the number of involved samples and filtered job postings of the top 1000 companies. After the above preprocessing, we got 215,308 job postings. Based on the observable contexts, we extracted continuous and discrete features to form the input of the model. The detailed descriptions of feature extraction can be found in Supplementary Table S4.

**Overall process**. The pseudocode of the overall model training and applying process of this paper can be found in Algorithm 1.

**Algorithm 1**. Overall process

**Require:** $D_{train}$: training set; $D_{test}$: testing set; $\eta$: learning rate; *MaxIter*: the number of training iterations.
1: Build model $\mathcal{M}$ with initial parameter $\Phi$;
2: /\*\*Training\*\*/
3: **For** $it \in 1 \cdots MaxIter$ **do**
4: $S_{batch} \leftarrow$ randomly split $D_{train}$ into batches;
5: **For** each $D_{batch} \in S_{batch}$ **do**
6: $d\Phi = 0$;
7: **For** each $(skillset, context, y) \in D_{batch}$ **do**
8: $d\Phi = d\Phi + \frac{\partial Loss(\mathcal{M}(skillset, context; \Phi), y)}{\partial \Phi}$;
9: $\Phi = \Phi - \eta d\Phi$;
10: /\*\*Validation\*\*/
11: $Y_{pred}, Y_{true} \leftarrow$ empty lists;
12: **for** each $(skillset, context, y) \in D_{test}$ **do**
13: Predict salary range $\widetilde{y} = \mathcal{M}(skillset, context; \Phi)$;
14: Store $\widetilde{y}$ in $\mathbf{Y}_{pred}$;
15: Store $y$ in $\mathbf{Y}_{true}$;
16: Calculate $MAE(\mathbf{Y}_{pred}, \mathbf{Y}_{true})$ and $RMSE(\mathbf{Y}_{pred}, \mathbf{Y}_{true})$;
17: /\*\*Value Estimation\*\*/
18: **For** $(skillset, context, y) \in \mathbf{D}_{train} \cup \mathbf{D}_{test}$ **do**
19: **For** each $(level, skill) \in skillset$ **do**
20: Estimate value $v$ and domination $d$ for $(level, skill, context)$ with $\mathcal{M}(\cdot; \Phi)$.
21: Analyze skill value;

### Context-aware skill valuation network

*Temporal skill embedding*. Considering that the skills' traits change over time, CSVN assigns temporal embeddings for skills at each time interval. To reduce model complexity, we use the idea of Matrix Factorization[23] and assume the skill embedding is composed of a low-ranked embedding and a latent projecting matrix. Formally, $\mathbf{E_s^{(t)}} = (\mathbf{W^{us}})^{(t)}\mathbf{W^{vs}}, \quad t = 1, 2, \cdots, T$, where $\mathbf{E_s^{(t)}} \in \mathbb{R}^{N_s \times de}$ stores the skill embeddings of the *t-th* time interval, $T$ is the number of time intervals, $N_s$ denotes the size of the skill vocabulary, $(\mathbf{W^{us}})^{(t)} \in \mathbb{R}^{N_s \times dl}$ is the low-ranked skill embeddings of the *t-th* time interval, $\mathbf{W^{vs}} \in \mathbb{R}^{dl \times de}$ is the latent projection shared by all the time intervals, de is the embedding dimension, dl is the number of latent factors. Though the temporal embedding gives CSVN the ability to model skills' dynamic changes, it brings higher model complexity. To avoid over-fitting, we add a temporal regularization to the model, formulated as

$$L_t = \sum_{t=1}^{T-1} \| (\mathbf{E^s})^{(t+1)} - (\mathbf{E^s})^{(t)} \|_F, \tag{1}$$

where $\| \cdot \|_F$ denotes the Frobenious norm. $L_t$ constrains the temporal embeddings not to change sharply. With the temporal skill embedding, our model can distinguish the development and change on skill semantic over time and maintains low model complexity. However, it should also be noticed that our model is not a forecasting model as training data of each time period is needed to train the corresponding embedding.

*Skill-context interaction extraction*. To increase fitting ability, CSVN takes both continuous context vectors (e.g., salary statistics of a city) and discrete contexts (e.g., city index) as inputs. Then, inspired by the famous CTR prediction model DeepFM[24] in the field of recommender system, CSVN extracts both deep and shallow interactions between these job contexts and the skill. Specifically, the input contexts are processed in different manners and go though linear projection, multiplicative operation and Multi-Layer Perceptron (MLP) to extract interaction of different orders. Formally, each continuous context $i \in \mathcal{C}$ inputs a feature vector, written as $\mathbf{o_i^c} \in \mathbb{R}^{d_i}$, where $\mathcal{C}$ denotes continuous job contexts. Each discrete context $i \in \mathcal{D}$ inputs an index, CSVN encodes it into an one-hot representation $\mathbf{o_i^d} \in \mathbb{R}^{m_i}$, where $m_i$ is the maximum possible value of this context. Then the linear projection extracts the first-order interaction as

$$\mathbf{h_1} = \sum_{i \in \mathcal{C}} \mathbf{W_i^{cl}} \mathbf{o_i^c} + \sum_{i \in \mathcal{D}} \mathbf{W_i^{dl}} \mathbf{o_i^d} + \mathbf{W^{sl}} \mathbf{e^s} + \mathbf{b^l}, \tag{2}$$

where $\mathbf{e^s} \in \mathbb{R}^{de}$ denotes the input skill's current embedding vector, $\mathbf{W_i^{cl}} \in \mathbb{R}^{do_1 \times d_i}$, $\mathbf{W_i^{dl}} \in \mathbb{R}^{do_1 \times m_i}$, $\mathbf{W_i^{sl}} \in \mathbb{R}^{do_1 \times de}$ and $\mathbf{b^l} \in \mathbb{R}^{do_1}$ are the trainable parameters, do$_1$ is the output dimension. Then, multiplicative operation extracts the second-order interactions. Specifically, each discrete context $i \in \mathcal{D}$ is first

assigned with an embedding $\mathbf{e_i^d} = \mathbf{o_i^d} \mathbf{W_i^e}$, where $\mathbf{W_i^e} \in \mathbb{R}^{m_i \times de}$ stores the value embeddings of context $i$. For continuous context $i \in \mathcal{C}$, we project the feature vector into the space of discrete job contexts, written as $\mathbf{e_i^c} = \mathbf{o_i^c} \mathbf{W_i^p} + \mathbf{b^p}$, where $\mathbf{W_i^p} \in \mathbb{R}^{d_i \times de}$ and $\mathbf{b^p} \in \mathbb{R}^{de}$ are trainable parameters. The multiplicative operation is formulated as

$$\mathbf{h_2} = \sum_{i \in \mathcal{C}} \sum_{i \neq j, j \in \mathcal{C}} \mathbf{e_i^c} \odot \mathbf{e_j^c} + \sum_{i \in \mathcal{D}} \sum_{i \neq j, j \in \mathcal{D}} \mathbf{e_i^d} \odot \mathbf{e_j^d} + \sum_{i \in \mathcal{C}} \sum_{j \in \mathcal{D}} \mathbf{e_i^c} \odot \mathbf{e_j^d} + \mathbf{e^s} (\sum_{i \in \mathcal{C}} \mathbf{e_i^c} + \sum_{i \in \mathcal{D}} \mathbf{e_i^d}),$$

(3)

where $\odot$ denotes element-wise multiplication. At last, MLP extracts the higher order information, which is tiled by several fully connected layers, formulated as

$$\mathbf{x^{(0)}} = \mathbf{o_0^c} | \mathbf{o_1^c} | \cdots | \mathbf{e_0^d} | \mathbf{e_1^d} | \mathbf{e^s}, \quad \mathbf{x^{(k)}} = \sigma(\mathbf{x^{(k-1)}} (\mathbf{W^m})^{(k)}), \quad k = 1, 2, \cdots, K \quad (4)$$

where $K$ is the depth, $(\mathbf{W^m})^{(k)} \in \mathbb{R}^{d_m^{(k-1)} \times d_m^{(k)}}$ and $\mathbf{x^{(k)}}$ denotes the parameter and the output of the $k$-th layer, $\sigma$ denotes the activation function, $*|*$ denotes concatenating two vectors. We set the final output $\mathbf{x^{(K)}}$ as the high-order interaction $\mathbf{h_3}$.

To provide context-skill representation for domination modeling, this MLP has a multi-head structure. Specifically, since outputs of the shallow layers are general context-skill interactions, while the whole MLP extracts value related information, the output of some shallow middle layer is fed into ASDN to extract domination related features. The details will be described in the salary prediction part.

*Constrained value range modeling.* CSVN estimates the value range by predicting its bounds. To assure that the predicted bounds can form a meaningful value range, we have two constraints. First, since skill value is a special case of salary, its lower bound is non-negative. Second, the upper-bound value is no less than the lower-bound value. We concatenate the extracted interaction of different orders, then estimate the range with two constrained linear projection, formulated as

$$v^l = [\mathbf{h_1}|\mathbf{h_2}|\mathbf{h_3}] \mathbf{W^l} + b^l, \quad v^u = [\mathbf{h_1}|\mathbf{h_2}|\mathbf{h_3}] \mathbf{W^u} + b^u, \quad \text{s.t.} \quad 0 \leq v^l \leq v^u. \quad (5)$$

As $v^l$ and $v^u$ are intermediate variables of SSCN, its whole training process becomes a constrained optimization. However, it is hard for deep learning models to deal with constraints. Though we can add a soft constraint regularization to the loss function, it cannot guarantee the constraints are strictly satisfied and can easily cause the model fail to converge. To avoid constrained optimization and enable gradient descent, we adjust the network structure so that the constraints are naturally satisfied. Specifically, we add a non-negative activation to the lower-bound output, formulated as

$$v^l = \max([\mathbf{h_1}|\mathbf{h_2}|\mathbf{h_3}] \mathbf{W^l} + b^l, 0). \quad (6)$$

Next, instead of directly predicting the upper-bound value, we change the mission of the second linear projection to output the gap $p$ between the bounds, the upper bound is thus calculated as $v^u = v^l + p$. The upper bound is guaranteed to be no smaller than the lower bound if we constrain the gap to be non-negative, formulated as $p = \max([\mathbf{h_1}|\mathbf{h_2}|\mathbf{h_3}] \mathbf{W^g} + b^g, 0)$.

**Attentive skill domination network.** In Fig. 2d, we show the structure of ASDN. ASDN use features extracted by CSVN as the input, denoted by $\mathbf{IA}$. From $\mathbf{IA}$, it first independently extracts two kinds of skill representations with MLP. ASDN first extracts an important representation for each skill, which implicates the traits of the skill that impact their domination, e.g., some skills may be common and easy to become the major part of the jobs. Meanwhile, ASDN extracts an influence representation for each skill to model their influence on domination to each other. We use $\mathbf{X_{imp}} \in \mathbb{R}^{N \times dp}$ and $\mathbf{X_{inf}} \in \mathbb{R}^{N \times di}$ to denote the importance/influence representation, where each row of them is a skill's representation and $N$ denotes the number of appeared skills.

ASDN supposes the domination of skill is affected by three factors, which are its own importance, the global influence from all the skills, and the local influence from the related skills. The global influence is calculated as the averaged influence vector of all the skills, written as $\mathbf{Q} = \frac{\mathbb{1}^\top \mathbf{X_{inf}}}{N}$, where $\mathbb{1} \in \mathbb{R}^{N_s}$ is a vector whose elements are all 1. The global influence is the same for all the skills, we regard it as the query in the attention mechanism. To model the influence from the neighboring skills, we apply a simple Graph Convolutional Network (GCN)[25] on the skill graph to extract the local influence, formulated as

$$\mathbf{U^{(0)}} = \mathbf{X_{inf}}, \quad \mathbf{U^{(k)}} = \sigma(\mathbf{A} \mathbf{U^{(k-1)}} (\mathbf{W^g})^{(k)}), k = 1, 2, \cdots, K_c, \quad (7)$$

where $K_c$ is the depth of GCN, $\mathbf{A} \in \mathbb{R}^{N \times N}$ is the adjacency matrix of the skill graph, $\mathbf{A}_{i,j}$ denotes the edge weight from skill $i$ to skill $j$, $\mathbf{U^{(k)}} \in \mathbb{R}^{N \times d^{(l)}}$ stores the output vectors of all the nodes in the $k$-th layer, $d^{(k)}$ is the output dimension, and $\mathbf{W^g} \in \mathbb{R}^{d^{(k)} \times d^{(k+1)}}$ is the trainable parameter. We concatenate the importance vectors with the local influence vectors as the keys and calculates the dominations of each skill with an attention layer, formulated as

$$\tilde{\mathbf{a}} = \tanh\left(\mathbf{Q} \mathbf{W^q} + [\mathbf{U^{(K_c)}}|\mathbf{X_{imp}}] \mathbf{W^k}\right) \mathbf{W^v}, \quad \mathbf{a} = \text{softmax}(\tilde{\mathbf{a}}), \quad (8)$$

where $\mathbf{a} \in \mathbb{R}^N$, the element $\mathbf{a}_i$ represents domination of the $i$-th skill, $\mathbf{W^q} \in$

**Table 4 The network configurations.**

| Name | Value | Name | Value |
|---|---|---|---|
| Embedding Size | 16 | Latent Factor Size | 6 |
| CSVN MLP | [64, 64, 64, 16, 16, 16] | Shared Depth | 3 |
| Influence MLP | [16, 16, 16] | GCN layers | [16, 16] |
| Importance MLP | [16, 16, 16] | | |

$\mathbb{R}^{di \times da}, \mathbf{W^k} \in \mathbb{R}^{(d^{(k)} + dp) \times da}$ and $\mathbf{W^v} \in \mathbb{R}^{da}$ are the trainable parameters. To guarantee that each skill has separate domination factors for lower-bound and upper-bound salary, ASDN trains two sets of the above attentional parameters.

**Job salary prediction.** For a job posting $\mathbf{J_j}$, SSCN models its job salary as the weighted average of the skill value. The lower bound salary $\tilde{y}_j^l$ and upper bound salary $\tilde{y}_j^u$ is estimated as

$$\tilde{y}_j^l = \sum_i^{|\mathbf{S_j}|} (v^l)_j^{(i)} (a^l)_j^{(i)} \quad \tilde{y}_j^u = \sum_i^{|\mathbf{S_j}|} (v^u)_j^{(i)} (a^u)_j^{(i)}, \quad (9)$$

where $(v^*)_j^i$ represents the value bound of the $i$-th skill in $\mathbf{J_j}$, $(a^*)_j^i$ represents the corresponding domination factor. We set the loss function to be the difference between the predicted and the real salary bounds, formulated as

$$L_s = \frac{\lambda_l}{|\mathcal{J}|} \sum_j^{|\mathcal{J}|} (\tilde{y}_j^l - y_j^l)^2 + \frac{\lambda_u}{|\mathcal{J}|} \sum_j^{|\mathcal{J}|} (\tilde{y}_j^u - y_j^u)^2, \quad (10)$$

where $y_j^*$ denote the observation of job salary bounds, $\lambda_l$ and $\lambda_u$ are hyper-parameters for balancing the importance of these two loss, $|\mathcal{J}|$ denotes the job postings set.

Combining the $L_s$ with the skills' temporal regularizer $L_t$, we formulate the loss function of SSCN as

$$L = \frac{\lambda_l}{|\mathcal{J}|} \sum_j (\tilde{y}_j^l - y_j^l)^2 + \frac{\lambda_u}{|\mathcal{J}|} \sum_j (\tilde{y}_j^u - y_j^u)^2 + \beta \sum_{t=1}^{T-1} \| \mathbf{E_s^{(t+1)}} - \mathbf{E_s^{(t)}} \|_F, \quad (11)$$

where $\beta$ is a hyperparameter balancing the importance of the temporal regularizer.

**Network configuration.** The network configurations can be found in Table 4. Since the lower-bound salary is smaller than the upper bound, we set $\lambda_l$ and $\lambda_u$ to be 2 and 1. The time regularizer $\beta$ was set to be 0.004. We use residual structure[26] to accelerate the training and Leaky ReLU[27] as the activation function. The weights are initialized with glorot normal initializer[28]. For optimization, we use Adam optimizer[29]. We found slight changes in parameters did not affect much on the performance. Specifically, the additional parameter experiments can be found in Supplementary Information.

**Baseline methods for salary prediction.** Our baseline methods for salary prediction including four parts:

- Classic regression models including linear regression (LR), Support Vector Machine (SVM), and Gradient Boosting Decision Tree (GBDT). Since these methods process the structured feature vectors of fixed size, we concatenated the one-hot skillset representation, the averaged features of skills, and job context as their input.
- Deep Neural Network with the same depth and a similar number of variables as SSCN for fairness of comparison. The input was also the concatenated feature vector.
- Holistic Salary Benchmarking Matrix Factorization (HSBMF)[30]. HSBMF is the state-of-the-art salary benchmarking model. HSBMF groups the job advertisements into posts and predict their salary with matrix factorization. We used the job contextual information and skill requirements for building regularization matrices in HSBMF to assure it considers the same information as SSCN.
- State-of-the-art text mining-based methods. We compared two groups of typical methods that model the job postings as texts. The first group consists of well-adopted Natural Language Processing (NLP) network architectures trained in an end-to-end manner with our data, including Convolutional Neural Network (TextCNN)[31,32], Hierarchical Attention Network (HAN)[33], and the recently proposed Transformer-XL[34]. In these models, we used pre-trained Chinese word embeddings[35] to initialize the parameters. The second group consists of state-of-the-art pre-trained models, including Bidirectional Encoder Representations from Transformers (BERT)[36], Robustly optimized BERT approach (RoBERTa)[37], and XLNet[38]. To better process our input data, we have adopted models trained with Chinese corpus[39].

We also disabled some parts of SSCN to show their effectiveness, including two parts:

- "CSVN + Mean", where we replaced ASDN with a mean pooling layer.
- "SSCN (Independ)", where we disabled the range prediction part and train the models for the upper bound and lower bound independently.

For all the compared methods that are not designed for range prediction, we separately train the lower-bound and upper-bound regression model with them and validate their performances independently.

**Validation**. We repeated 10 times of hold-out validation on the models. Specifically, at each time, we randomly split the data into training and testing set with a ratio of 4:1. We used the training data for model training and used the testing data for performance evaluation.

**Reporting summary**. Further information on research design is available in the Nature Research Reporting Summary linked to this article.

## Data availability

The job posting data that support the findings of this study are available in figshare with the identifier "10.6084/m9.figshare.14060498"[40]. All data generated or analyzed during this study are included in this published article (and its Supplementary information files). Source data are provided with this paper.

## Code availability

Codes of this paper are available in CodeOcean with the identifier "10.24433/CO.0239280.v1"[41].

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

## Acknowledgements

We thank the members of the Baidu Talent Intelligence Center for their support, ideas, and encouragement. The research work supported by the National Key Research and Development Program of China (Grant No. 2018YFB1004300), the National Natural Science Foundation of China (Grant Nos. U1836206, U1811461, 61773361, 91746301, and 61836013), the Project of Youth Innovation Promotion Association CAS (Grant No. 2017146).

## Author contributions

This work was accomplished when Y.S. and Q.Z. working as interns in Baidu supervised by H.S.Z. H.S.Z. came up with the idea of market-oriented skill valuation. Y.S. and H.S.Z.

formulated the problem of *Salary-Skill Value Composition Problem*. Y.S. designed and implemented *Salary-Skill Composition Network* under the guidance of F.Z.Z. and H.S.Z. Q.Z. gave important advice on model structure. Y.S. and Q.Z. processed the data. Y.S., F.Z.Z., and H.S.Z. conceived the experiments and evaluated the results. F.Z.Z., H.S.Z., Q.H. and H.X. advised on the literature review, data process and technical design of this work. Y.S., H.S.Z. and H.X. wrote the paper. H.S.Z., F.Z.Z. and H.X. managed this project.

## Competing interests
H.S.Z. is currently affiliated with Baidu. Y.S. and Q.Z. are currently affiliated with Baidu as research interns. The other authors declare no competing interests.
