## [Peer Review File · Nature Communications]

Reviewers' Comments:

Reviewer #1:

Remarks to the Author:

Thank you for giving me the opportunity to review the manuscript entitled "Market-oriented Job Skill Valuation with Cooperative Composition Neural Network." In their work, authors aimed at introducing a new way of value assessment of job skills. More specifically, authors analysed job adverts and the profiles defined therein. They subsequently used machine learning algorithms to come up with a model that can predict the value of different job skills defined as their direct effect on salary when taking the context of an open position into account.

Overall, I enjoyed reading this manuscript and I believe that what authors present might indeed be an innovative and interesting approach. I am happy to say that there are many things I liked about the manuscript. At the same time, I noted a number of shortcomings, some of them essential, and I think major additions, changes, and amendments are needed for this manuscript to become publishable in such a high-ranking journal. In this, I also want to be very transparent that at this point I am unable to make a final judgment whether I will be able to recommend this manuscript for publication even after substantial revisions, but I do believe in the potential of the current submissions and would suggest that authors revise the manuscript.

I have several overarching comments and concerns, which I will mention first. I will then proceed with providing more detailed feedback that adds specifics to the overarching comments. I hope my comments are helpful in revising the manuscript. I would like to stress again that I have the highest respect for the authors' work.

I would also like to disclose that I am no expert in the methods that the authors apply so please consider this when reading my methodological comments (in fact, my background is not in economics, but in psychology). I hope that one of the other referees can give competent feedback on the specific methods that were employed by the authors.

Overarching comments:

- a) Many of the core theoretical concepts that the authors investigate in this paper lack a clear definition. In fact, in my reading several of the concepts such as job skills or value are operationally defined. However, the paper would be much stronger with a theoretical definition of the concepts and a straightforward rationale for the chosen operationalisations. I was also under the impression that the authors made frequent use of buzz words that have little theoretical value;
- b) The general rationale underlying this paper is, in a way, rather simplistic in the sense that it assumes that context and job skills directly (and linearly?) influence salary as the core outcome variable. While all models are a simplification of a complex reality, a stronger rationale is needed here why the specific model that was chosen by authors is a valid reflection of reality;
- c) Authors present many descriptive (and graphic) analyses. This is illustrative in many places but provides little basis for a straightforward interpretation of the results. This should be remedied through amending numerical information in a potential revision of the manuscript;
- d) I can see the relevance of this paper for a targeted audience and I am sure that many readers from the field of economics will be interested in this paper. I am missing the broader rationale on why and how this is relevant for a general audience;
- e) Fit to Nature Communications: I see this paper clearly located in the field of economics but economics is not explicitly identified as within the scope of the journal on the journal's website. I think it should be an editorial decision whether this paper is in the scope of Nature Communications, but I wanted to flag this in my review.

Along the lines of these overarching points, there are a number of more specific points that should be addressed in a revision:

1. Overall, the manuscript is well written, but there are a few typos and grammatical errors. Please carefully check language and grammar;

2. As mentioned above, there is a rather extensive use of buzz words such as salary structure, skill gap, technological changes, talent (and others) without a clear definition of these concepts. It is not clear what is meant by them and how they relate to each other. Please drop or provide definitions of all concepts that you use;
3. On a related note, the paper lacks a definition of job skill even though this is a crucial concept for the entire article; similarly, the relation between job skill and salary is not clearly elaborated upon either. I urge authors to add additional theoretical background to the core concepts that they mention in their article;
4. The underlying assumption that – given that the context is constant – the salary is a direct function of the skill set is not sufficiently reasoned for even though this is a core assumption. I agree that this assumption is reasonable and aligns with common sense, but in such a paper I would expect to learn more about the core assumptions of the underlying model (and its limitations);
5. The authors should provide more information on existing models to predict salary and they should elaborate what the specific advantage / strengths of their model is;
6. Figure 1: It was not clear to me what the figure is meant to depict;
7. The authors try to provide new definitions of complex constructs, for instance by using salary as outcome. They claim that this approach is new and has not been investigated before; however, on the basis of their description, I did not fully understand what the specific operationalisations of the different concepts were. For instance, how are “job information other than skill requirement” or “context” defined and, subsequently, operationalized?
8. Along several of the previous comments, I will add as a core point that there is a difference between a theoretical concept (e.g., job value) and specific operationalisations (e.g., salary). I see it as a weakness of the current version of the manuscript that it does not make this distinction and confuses the two;
9. With regard to the empirical approach, how did authors make sure that they would capture all relevant job postings from different job markets? The type of adverts that they investigate might be very much restricted to white-collar jobs of rather highly skilled individuals. This, in turn, raises the question of generalizability. How about different languages and different countries? My understanding is that authors target a very specific (albeit large) job market and that leads to the question whether these results are likely to generalize across other (job) areas;
10. How are the interactions of different skills considered? For instance, being highly proficient in 6 languages might increase the salary more than twice as much as the increase for being highly proficient in 3 languages. Does the model account for such relations or is the model additive in nature (which would probably be too simple)? And, in addition to this, would there be enough data to estimate these interaction effects?
11. Along the previous point, authors should report on alternative models with alternative parameters in the supplementary material to show that their model generalizes and is not a specific solution with little transfer value;
12. How does the model perform in predicting future developments? As time is explicitly considered in the model, authors should say something on forecasting;
13. Table 1 left me a bit puzzled. What scale is “value”? Also, I am not clear how it makes sense that a higher level of a skill leads to a decrease in value. This requires careful explanation and consideration. It might indicate that the model does not really work because it cannot take into account complex interaction effects (e.g., some skills occurring only in the context of other skills and then, in this specific context, being less valuable) or there are suppression effects at work. In any case, if I am not mistaken in my interpretation of Table 1, then I am concerned about the highly counterintuitive finding that a higher skill level leads to lower salary and it would be important to get to the bottom of this (might be a method effect, which then relates to the question of generalizability);
14. As authors mention throughout the manuscript, context is a relevant concept. Could authors provide additional information on how they measured context?
15. Would it be possible for authors to report more long-term data (e.g., from the last 10 or 20 years)? This could give interesting insights on general developments on the relevance of different skills for different markets;

16. Many of the results are interesting, for instance the ones in Figure 3. However, I noted that many of the interpretations rely on analyzing descriptive patterns in the data by the mere looks of it. This comes along with a high chance of capitalizing on random trends and findings (albeit the large data set) and of being inaccurate. Maybe authors could choose a more formal approach throughout the manuscript to test their assumptions?

17. I like the notion that many jobs require generic skills and that this type of skill set applies more broadly to different jobs as distinct and rather narrow skills. This could be made stronger in the manuscript and could be connected to the vast literature on 21st century society, educational changes in a digital world, and how the employment market has been changing over the last decades;

18. Figure 4 provides nice illustrations but is very descriptive. Could the information contained in this figure be calculated/presented using actual statistics?

19. Table 3: I am not clear about the scale for columns 2 and 3;

20. The next step in the research presented by the authors would be to have a model that gives an estimation of a fair salary (both for employer and employee) and also provides guidance for applicants on where to apply given their job profile. Authors do mention this as a possibility, but I wonder whether it wouldn't be possible to present such an equation. In any case, authors should discuss the relevance of such a model and of their approach in way that it takes into account broad potential applications;

21. Case study: I have doubts that this can be applied to individual job postings. Authors should distinguish between averaged postings in the model and applying it to individual cases.

Reviewer #2:

Remarks to the Author:

This paper addresses the problem of evaluating and predicting the value of a skill as it emerges by processing and analysing online job postings (aka, online job vacancies/advertisements). The problem is relevant and has a practical significance for the industry, the academy, and for the governments, too. The paper is well written and contextualised. However, I see two main issues related to (i) significance of the results and (ii) technical Contribution (see comments below).

MAJOR:

[Motivation and Impact]

-- The authors seem to do not be familiar with the domain of Labour Market Intelligence, that employs AI algorithms to derive knowledge useful for understanding labour market dynamics and trends. They do not discuss the relevance and impact of processing vacancies for understanding labour market dynamics indeed (see, e.g.: Frey, C.B., Osborne, M.A.: The future of employment: How susceptible are jobs to computerisation? *Technological Forecasting and Social Change* 114(Supplement C), 254 – 280 (2017) ; Alabdulkareem, A., Frank, M.R., Sun, L., AlShebli, B., Hidalgo, C., Rahwan, I.: Unpacking the polarisation of workplace skills. *Science advances* 4(7) (2018))

as well as the importance for governments given last relevant projects on collecting and processing job posting skills (see, e.g. <https://www.cedefop.europa.eu/it/about-edefop/public-procurement/real-time-labour-market-information-skill-requirements-setting-eu> ; https://ec.europa.eu/eurostat/cros/content/essnet-big-data_en ;). Furthermore, some recent and relevant works are omitted. Specifically, the authors missed comparing with a very similar approach that employs vector-space models for salary calibration-prediction (Zhang, D., Liu, J., Zhu, H., Liu, Y., Wang, L., Wang, P., Xiong, H.: Job2vec: Job title benchmarking with collective multi-view representation learning. In: *CIKM*. pp. 2763–2771 (2019))

[Contribution]

-- The authors do not discuss at all *how* information that they use in their model are extracted from job postings (aka, online job advertisements). They just mention the online source from

which they have been collected, without providing any information about the quality (update time, missing values, num of duplicated vacancies, coverage over all occupation types, etc.)

Indeed, the problem of extracting such information such as (i) company name (ii) location, (iii) salary is far from straightforward, and it mainly affects the trustability of the analyses based on that information.

-- Authors say they trained their model on a "large-scale" set of job postings composed by 800k vacancies over 36 months. However, those publication rate is shallow compared with the publication rate of the major EU websites (see <https://www.cedefop.europa.eu/en/data-visualisations/skills-online-vacancies>) and this limits both the effectiveness of your approach and the significance of the results (intended as the value of skills they aim to assess).

Furthermore, they do not discuss in any way.

(i) the type of online source, if you include aggregators, the number of duplicate vacancies grows a lot (up to 30% according to a recent study), how did you identify duplicate vacancies?

(ii) empirically, the salary information (that is crucial in your model) is present in only 15-20% of vacancies. Which is the distribution of salary variable within vacancies? how has this information been collected?

(iii) For comparing and analysing vacancies, the most important step is to classify vacancies over standard occupation taxonomies (e.g., ISCO/ESCO for Europe, O*Net for US, see e.g. Boselli, Cesarini, Mercurio, Mezzanzanica: Classifying online Job Advertisements through Machine Learning. *Future Gener. Comput. Syst.* 86: 319-328 (2018)). This is a crucial to understand if analyses (here, the Salary-Skill Value Composition analysis) is uniformly distributed over sectors/occupations, as well as to assess if your ads set is overestimating/underestimating some occupations/sectors. Otherwise, the analyses represent a technical exercise that does not give significant benefit to the user as it cannot explain the reality.

[technical Contribution]

The problem definition is intuitive, and the article is undoubtedly well written. However, I am ambivalent about the strength of its theoretical Contribution. The proposed method for learning skill value is a straightforward application of existing work. The Contribution is thus limited to the use of such techniques in a new domain and task. Even though the use of attention and cooperative networks in Labour Market is quite new, this is an application paper without major technical novelty.

Experiments are not robust. Hyperparameter tuning for the author's model and baseline models is missing, and so validation strategy (cross-validation, holdout, how overfitting is missing, etc.), weights initialisation, etc. Those are usually essential details when dealing with neural networks, especially the complex model presented by the authors.

-- Related works about collaborative neural networks and attention networks are missing.

MINOR:

typos and unclear passages:

line 2: on the part of both talents

line 5: help companies to delicately design

line 16, 17: the most critical issue of prohibiting the 16 skill valuation is that there usually lacks of labeled skill value

line 62: we expect the function g to be interpretable that can explicitly reveal etc.

Summary of Revisions

Title: Market-oriented Job Skill Valuation with Cooperative Composition
Neural Network

Submission ID: NCOMMS-20-27613d

Authors: Ying Sun, Fuzhen Zhuang, Hengshu Zhu, Qi Zhang, Qing He, and
Hui Xiong

First of all, we would like to thank the editor and two anonymous reviewers for their valuable comments. Their constructive suggestions have not only helped for the improvement of this paper in a number of ways, but also given a lot of inspirations for our future research works. In the following, we first summarize the major revisions of this paper, and then address the specific comments from the two reviewers. As a convention, the “last submitted version” refers to the version we submitted in July 2020 for initial review, and the “revised version” refers to the current submitted version.

1. Revision Summary

We have revised the paper according to the comments from the editor and two reviewers. The revised details have been summarized as follows.

In the revised version, we have provided substantial new content:

- We have provided substantial supplementary information for improving the understanding of this paper and further validating our research findings. Specifically, we have added detailed literature review, the clear definitions of terms and concepts, raw data analysis, technical details, numerical statistics, skill value analysis, and more experiments.
- We have collected a new dataset of designer-related job postings and conducted extensive experiments to demonstrate the generalization of the proposed model.
- We have provided some discussions for showing the potential applications of our work and the limitations of this study.

In the revised version, we have also revised some content:

- First, we have revised the descriptions of research background and methodology for further clarifying the motivation, the contributions, and the significance of this work.
- Also, we have enriched the experimental results by adding more empirical analysis, validations and discussions, which are consistent with the original results but more informative.
- Finally, we have revised some descriptions, formats and texts for improving the readability of this paper. Also, we carefully polished the use of language. For example, we corrected typos, grammar errors and confusing expressions in the revised version.

2. Response to Reviewer 1

Comment 1: *Overall, I enjoyed reading this manuscript and I believe that what authors present might indeed be an innovative and interesting approach. I am happy to say that there are many things I liked about the manuscript. At the same time, I noted a number of shortcomings, some of them essential, and I think major additions, changes, and amendments are needed for this manuscript to become publishable in such a high-ranking journal.*

Response: We very much appreciate your encouraging words and recognition on the novelty of our approach. Meanwhile, according to your constructive comments and suggestions, we have thoroughly revised this paper in a number of ways. In the following, we will introduce the details of our revisions.

Comment 2: *Many of the core theoretical concepts that the authors investigate in this paper lack a clear definition. In fact, in my reading several of the concepts such as job skills or value are operationally defined. However, the paper would be much stronger with a theoretical definition of the concepts and a straightforward rationale for the chosen operationalisations. I was also under the impression that the authors made frequent use of buzz words that have little theoretical value;*

Response: Many thanks for your valuable comments and sorry for the confusion! We fully agree with you that a theoretical definition of the concepts will make skill value modeling more rigorous and comprehensive. Indeed, some previous works have revealed that job skills together with other factors have explicit

impact on the job salary [1,2,3]. However, to the best of our knowledge, there still lacks a complete theoretical framework for assessing the market-oriented value of job skills. Therefore, in this paper, we studied the problem of market-oriented job skill valuation from a data-driven perspective. To this end, instead of using a top-down theoretical manner for investigating the concepts, we approach the problem by exploiting the features extracted from the data, such as skill words, job salary range, and contextual features. Specifically, as a convention of data mining practice [4, 5, 6], in this paper, we focus on building a machine learning model to automatically learn the quantitative relationships between job skills and job salaries observed from the large-scale job advertisement data. We believe this work can provide a novel data-driven perspective for addressing the challenges in job skill valuation.

According to your valuable suggestions, we have made the following revisions. First, to facilitate the understanding of our data-driven approach for job skill valuation, we have given more explicit descriptions of the concepts and observations of our model, and provided more supplementary information. Specifically, “Supplementary Note 2: Data Preprocessing” introduces the details of our dataset, the data preprocess for extracting model observations (e.g., job skill, job salary, and contextual information), and the details of model features (see Supplementary Table S4). Second, as for the “buzz words”, we have appended a detailed list (see Supplementary Table S1) for explaining their definitions and the corresponding references.

Comment 3: *The general rationale underlying this paper is, in a way, rather simplistic in the sense that it assumes that context and job skills directly (and linearly?) influence salary as the core outcome variable. While all models are a simplification of a complex reality, a stronger rationale is needed here why the specific model that was chosen by authors is a valid reflection of reality;*

Response: Many thanks for your comments! Indeed, we fully agree that the relationship among job skill, salary and other factors is very complicated to model. Therefore, in this work, to achieve the goal of job skill value assessment, we design an explainable machine learning approach for modeling their relationships based on the observations obtained from job advertisement data, where all factors that influence the job salary other than job skills are regarded as the “context” and the job skill value is assumed to be context-aware. In this way, we only have to pay attention on the input (i.e., context and job skills) and output (i.e., job salary and skill value), while other latent influencing factors and relationships will be learned by the deep learning model.

In particular, inspired by the widely-adopted linear regression models for analyzing feature importance, we simplify the last layer of our deep learning model (i.e., modeling the predictive relationship between job skills and job salary) into a linear form. In this way, the job skill value can be explicitly connected to job salary, which also assures that the influence of individual skills on job salary can be explicitly measured. Meanwhile, another advantage of this setting is that the value of different job skills is independent and comparable with respect to the given job context.

Nevertheless, even if the last layer of our model is in the linear form, the whole structure of our model is non-linear and rather complicated, which can implicitly model the latent influencing factors for the final estimation. Specifically, it contains complicated structures such as Multi-Layer Perceptron, embedding layers, Factorization Machine, Graph Convolutional Network and attention layer, which can effectively model the complicated interaction among skills and contexts and model latent factors in an implicit manner. For example, while we cannot directly define the complex relationships among job skills when calculating their value, we let the model automatically learn these relationships based on a skill graph during the calculation of skill domination, which is also an important part for salary prediction. Indeed, based on the experimental results (e.g., Table 2), our model can significantly outperform all the baseline models, especially the traditional linear ones (e.g., SVM and LR), in terms of modeling job skill-salary relationship.

Finally, according to your comments, in the revised version, we have enriched the description of the technical motivation behind our model, and the discussion of the performance improvement of our model compared with the linear ones.

Specifically, we have rewritten the following paragraph in the Salary-Skill Composition Network Section to clarify the motivation:

In the real-world working scenario, the employees allocate their time and effort among the skills according to the importance of different job duties. In this way, the skill value is comparable and independent with each other. Considering that skills may have combinatorial influences on salary, we let the model catch skill interactions through modeling the domination. Specifically, the skill co-appearance is considered to influence the domination of each skill, which assures the model to peel explainable skill value that is only context-dependent while maintaining the model's fitting ability to general job postings.

Then, we added the comparison details with linear models in “Evaluation on salary prediction”:

Second, SSCN outperforms the linear models (i.e., SVM and LR). To assure the physical meanings of the skill value, SSCN simplifies the last layer of skill composition into a linear form. However, SSCN is still a complicated non-linear deep learning model that can seize the complicated relation between skill, context and salary. So it performs much better than the real linear models.

Comment 4: *Authors present many descriptive (and graphic) analyses. This is illustrative in many places but provides little basis for a straightforward interpretation of the results. This should be remedied through amending numerical information in a potential revision of the manuscript;*

Response: Thank you very much for your suggestion! In the revised version, we have revised the results of descriptive and illustrative analysis, and introduced more basis for a straightforward interpretation. In particular, according to your suggestions, we have introduced more numerical information in corresponding content, and provided all the numerical data related to the figures in the supplementary information (see “Supplementary Tables: Numerical Statistics of the Original Figures”).

Comment 5: *I can see the relevance of this paper for a targeted audience and I am sure that many readers from the field of economics will be interested in this paper. I am missing the broader rationale on why and how this is relevant for a general audience;*

Response: Many thanks for your comments and sorry for the confusion. Indeed, in this paper, we study a cross-disciplinary social economic problem; that is, how to assess the value of job skills with data-driven machine learning techniques from a market-oriented perspective. From the application view, we believe this study can benefit a variety of applications, such as talent recruitment, talent development, student education, knowledge management, business market analysis, etc. From the technical view, our model uses job salary data as indirect supervised information when the ground truth of skill value is unavailable. This provides an alternative solution for knowledge discovery from data with indirect supervision. According to your comments, in the revised version, we have further clarified the significance and relevance of this paper for general audiences.

Specifically, we have refined the descriptions on the significance of this study in “Introduction” Section. Meanwhile, we also discussed the potential applications in the “Discussion” Section.

Comment 6: *Fit to Nature Communications: I see this paper clearly located in the field of economics but economics is not explicitly identified as within the scope of the journal on the journal's website. I think it should be an editorial decision whether this paper is in the scope of Nature Communications, but I wanted to flag this in my review.*

Response: Thanks for the comment! Through the communication with the editor of this paper, we have the clear feedback that this paper is entirely within the scope of Nature Communications.

Comment 7: *Overall, the manuscript is well written, but there are a few typos and grammatical errors. Please carefully check language and grammar;*

Response: In the revised version, we have carefully polished the use of language by correcting typos, grammar errors and confusing expressions.

Comment 8: *As mentioned above, there is a rather extensive use of buzz words such as salary structure, skill gap, technological changes, talent (and others) without a clear definition of these concepts. It is not clear what is meant by them and how they relate to each other. Please drop or provide definitions of all concepts that you use;*

Response: Sorry for the confusion. In the revised version, we have introduced a detailed list (see Supplementary Table S1) for explaining the definitions of these terms and how they relate to each other.

Comment 9: *On a related note, the paper lacks a definition of job skill even though this is a crucial concept for the entire article; similarly, the relation between job skill and salary is not clearly elaborated upon either. I urge authors to add additional theoretical background to the core concepts that they mention in their article;*

Response: Many thanks for your valuable comments and sorry for the confusion. Actually, in this paper, we directly model the skills observed from the job postings. Specifically, we extracted the skill words from the job postings and transform each job description into a set of skill observations. Thank you for your advice on facilitating the rationale of our assumption on the relationship between skill and salary with theoretical basis. Indeed, abundant works have proved that skills have significant impact on job salary [1,2]. However, the relationship between skill and salary is complicated, and there still lacks an explicit and complete theoretical framework to quantify this relationship. Therefore, in this paper, instead of accurately modeling the relationship between salary and skills, we aim to provide an understandable and practical measurement of the influence of skills on salary by mining job salary data. Along this line, we consider the salary as the weighted sum of the skill value and the weights are computed based

on the skill co-appearance. In this way, we can obtain the estimation on skill value that is independent of the co-appearance of other skills. We agree that our measurement of skill value is far from perfect, but we believe it is an effective and practical measurement for quantifying the job skill value in a fine-grained way.

According to your suggestions, in the revised version, we have cited some related works about skill influence on salary in the “Introduction” Section and have described the skill extraction process in detail in the supplementary information (see “Skill and Level Extraction” in “Supplementary Note 2: Data Preprocessing”). Moreover, we have added explanations on “job skill” in Supplementary Table S1.

Comment 10: *The underlying assumption that – given that the context is constant – the salary is a direct function of the skill set is not sufficiently reasoned for even though this is a core assumption. I agree that this assumption is reasonable and aligns with common sense, but in such a paper I would expect to learn more about the core assumptions of the underlying model (and its limitations);*

Response: Thank you very much for your valuable comments. Indeed, we fully agree with you that the relationship among job skill, salary and other factors is very complicated to model. In this paper, to simplify the modeling process, we use the term “context” to represent all the factors other than job skills that may influence the job salary and assume skill value to be context aware. Meanwhile, similar to other relevant studies [7,8], we mainly focus on modeling the typical contexts that can be observed from the data. To achieve the goal of job skill value assessment, we design an explainable machine learning approach for modeling their relationship based on the observations of job advertisement data. In this way, we only need to pay attention on the input (i.e., context and job skills) and output (i.e., job salary and skill value), while other latent influencing factors and relationships will be learned by the deep learning model. This makes the whole model easy to be operated and the skill value influenced by observable contexts can be explicitly estimated, which strongly supports further explainable analysis. Nevertheless, a disadvantage of our deep learning model is that all the influencing factors and their complicated relationships are implicitly modeled as a blackbox, which is hard to be interpreted in a theoretical way.

Finally, according to your comments, in the revised version, we have enriched the description of the motivation behind our model, and the discussion on the model limitations.

Specifically, in the Section of Salary-Skill Value Composition Problem, we have added the following descriptions to clarify the assumption:

It should be noticed that, although there might exist more complicated relationships among job skills, context and salary, in the problem formulation,

we only consider the skill value is context-aware and can be combined together in a linear way to reflect the job salary. In this way, our model can facilitate the measurement of the influence of contexts on individual skills as well as the influence of skills on job salary.

Then, in the Section of Salary-Skill Composition Network, we have added the following descriptions about the motivation on the network structure:

In this way, the skill value is comparable and independent with each other. Considering that skills may have combinatorial influences on salary, we let the model catch skill interactions through modeling the domination. Specifically, the skill co-appearance is considered to influence the domination of each skill, which assures the model to peel explainable skill value that is only context-dependent while maintaining the model's fitting ability to general job postings.

Finally, at the end of the Section of Salary-Skill Composition Network, we have added the following descriptions to summarize the advantages and limitations of the proposed model:

Indeed, SSCN models the relationship among skills, context and salary based on the observations of job advertisement data in an end-to-end manner. As a common issue of deep learning models, all the influencing factors and their complicated relationships are implicitly modeled as a blackbox, which is hard to be interpreted in a theoretical way. Nevertheless, it also brings the advantage that we only need to pay attention on the input (i.e., context and job skills) and output (i.e., job salary and skill value), while other latent influencing factors and relationships will be automatically learned by the hidden layers. In this way, the model is easy to be operated, and meanwhile, the skill value influenced by observable contexts can be explicitly estimated, which strongly supports further explainable analysis.

Comment 11: *The authors should provide more information on existing models to predict salary and they should elaborate what the specific advantage / strengths of their model is;*

Response: Many thanks for your constructive suggestions. In the revised version, we have added a new section in the supplementary information (see “Data-driven Salary Prediction” in “Supplementary Note 1: Related Works”) to introduce the state-of-the-art literatures related to salary prediction and discussed their advantages and differences compared with our model.

Comment 12: *Figure 1: It was not clear to me what the figure is meant to depict;*

Response: Sorry for the unclear illustration. Indeed, the main idea of this paper is to train a skill valuation model (the Main Task) with the cooperation of salary

prediction (the Cooperative Task). Specifically, we want to train a skill valuation model with machine learning technology. However, under the supervised machine learning paradigm, we are supposed to train the model with a set of input-observation pairs. In this way, the model can learn a function that maps the input to the observation. We call the observation as “supervision” to the model as it gives feedback on the model prediction. To achieve this process, we are supposed to prepare a set of labeled samples formed as (context, skill, value), which is unavailable since the value of the skills are unknown. In this case, the task of skill valuation is called “unsupervised”. So we use salary prediction as a cooperation task to auxiliary the training of the skill valuation model. Specifically, salary prediction is a supervised task since we have labeled samples of (job, salary) in our data. We design a model that can simultaneously achieve skill valuation and salary prediction. SSCN predicts the skill value and composes skill value into job salary. Then the skill valuation part of the model can verify if it has estimated skill value properly by observing if the salary is predicted correctly. In this way, the supervised salary prediction task provides the unsupervised skill valuation task with indirect supervision.

According to your comments, in the revised version, we have modified Figure 1 in a more explicit presentation and revised corresponding descriptions to facilitate understanding.

Comment 13: *The authors try to provide new definitions of complex constructs, for instance by using salary as outcome. They claim that this approach is new and has not been investigated before; however, on the basis of their description, I did not fully understand what the specific operationalisations of the different concepts were. For instance, how are “job information other than skill requirement” or “context” defined and, subsequently, operationalized?*

Response: Sorry for the confusion. Actually, in this paper, we try to provide a data-driven solution to directly model the relation among observations in the job advertisement data. We use the word “context” to represent factors that may influence the job salary. We process the job advertisement and recognize information like city, company, time, which all act as contexts that influence skill value and job salary. Then we further measure a set of numerical features (for example, we count the number of job postings in a city as a feature of the city) to form the input of the model. In the revised version, for helping readers better understand our data-driven approaches, we have added more descriptions of data formation, data preprocessing and feature extraction in the supplementary information (see “Supplementary Note 2: Data Preprocessing”).

Comment 14: *Along several of the previous comments, I will add as a core point that there is a difference between a theoretical concept (e.g., job value) and specific*

operationalisations (e.g., salary). I see it as a weakness of the current version of the manuscript that it does not make this distinction and confuses the two;

Response: Thank you very much for pointing out this issue. Indeed, as mentioned before, we directly operated the observations (skill words, salary, company, city, etc.) and did not give explicit conceptual definitions in this data-driven work. Your comments do inspire us to model from the level of concepts in our future studies, which is difficult but no doubt significant. Thanks again.

Comment 15: *With regard to the empirical approach, how did authors make sure that they would capture all relevant job postings from different job markets? The type of adverts that they investigate might be very much restricted to white-collar jobs of rather highly skilled individuals. This, in turn, raises the question of generalizability. How about different languages and different countries? My understanding is that authors target a very specific (albeit large) job market and that leads to the question whether these results are likely to generalize across other (job) areas;*

Response: Many thanks for your valuable comments. Just as you mentioned, in this paper, we mainly focus on the white-collar jobs of highly skilled individuals, since the skills in these jobs will play an important role in salary setting. To be honest, we do hope that we could collect all relevant job postings from different job markets. However, data collection is a non-trivial task and we can only make it to collect a representative dataset from one of the largest and most popular Chinese online recruitment website of Internet-related industry. The dataset covers the IT industry of China and contains a large amount of job advertisements. Besides, it has high quality and contains abundant salary and contextual information. In this paper, we used this dataset for validation and the model turns out to be effective for estimating the skill value in Chinese job market. Actually, the model proposed in this paper is designed based on typical job advertisement data, which is generally applicable for recruitment data collected from other countries. Moreover, if job postings can be collected from all over the world, the country information can also be regarded as a contextual information input of our model. Therefore, the model can model and distinguish the skill value of different countries in a comparable manner. Although we do not have such available data currently, we hope our model could become an inspiration for researchers who have access to more abundant data sources.

Thank you again for your valuable suggestions. Accordingly, in the revised version, we have expanded a dataset consisting of designer-related job positions and conducted extensive experiments on it. The results can be found in the supplementary information as “Supplementary Experiment 3: Experiments on the Designer Dataset” Section.

Comment 16: *How are the interactions of different skills considered? For instance, being highly proficient in 6 languages might increase the salary more than twice as much as the increase for being highly proficient in 3 languages. Does the model account for such relations or is the model additive in nature (which would probably be too simple)? And, in addition to this, would there be enough data to estimate these interaction effects?*

Response: Sorry for the confusion. With the assumption that employees will allocate their time and energy to different skills during work, our model defines the salary as the weighted sum of the skill value. Indeed, this assures the estimated skill value to have explicit and comparable physical meaning. The model considers the skill interaction from the perspective of their influence on each other's importance (domination) in a job. Specifically, the model learns to model the skills' interaction on a skill graph, then accordingly estimates the domination of each involved skill.

Though the skill interaction is modeled for domination calculation, it is true that we still largely simplified the combinatorial effects of skills on salary, which may be extremely complicated in practice. For example, in addition to the case you have mentioned, I guess sometimes it is also possible that being highly proficient in 6 languages might increase the salary less than twice as much as the increase for being highly proficient in 3 languages. The reason is that the energy and time is restricted for a talent, so that the skills may have decreased edge effect.

In this paper, since we aim to learn skill value from data, we give a simplified assumption that is generally applicable to the job postings in the market, instead of trying to catch all the possible combinatorial effects. The special cases involving various combinatorial effects are regarded as outliers, which cannot prevent us from obtaining an explainable and general skill value measurement. Still, although we simplified the assumption, the performance of our model on salary prediction is still satisfied compared with other state-of-the-art baselines, which indicates that our assumption fits for the general case of the market.

As for the problem of more accurately estimating the interaction effects, I think this is an interesting but challenging topic. However, the number of skill combinations can be exponentially large. The skill interaction will differ for different combinations. As a result, it is extremely difficult to give an explicit, unified and easy-to-understand rule to comprehensively reveal the combinatorial skill impact. And even for estimating a single kind of combinatorial effect, I think it is still difficult to obtain enough data. Although the total number of job postings is large, the samples that can be used to analyze a specific kind of combination will be sparse. For example, to discover regular patterns of salary increase when adding skill requirements, we may need to collect a set of job pairs where the skill set of one job fully contains the skill set of the other. But the market job postings may contain few of these pairs.

According to your valuable comments, in the revised version, we have revised the descriptions in the section of “Salary-Skill Composition Network” and emphasized more on the motivations and details of the simplification on skill interaction, hoping which can better convey our ideas to the readers.

Specifically, we have rewritten the following paragraph in the “Salary-Skill Composition Network” Section to enhance the motivation:

In the real-world working scenario, the employees allocate their time and effort among the skills according to the importance of different job duties. In this way, the skill value is comparable and independent with each other. Considering that skills may have combinatorial influences on salary, we let the model catch skill interactions through modeling the domination. Specifically, the skill co-appearance is considered to influence the domination of each skill, which assures the model to peel explainable skill value that is only context-dependent while maintaining the model’s fitting ability to general job postings.

Comment 17: *Along the previous point, authors should report on alternative models with alternative parameters in the supplementary information to show that their model generalizes and is not a specific solution with little transfer value;*

Response: Thank you very much for your valuable suggestion. To validate the generalization of our model to other areas, in the revised version, we have expanded a dataset consisting of designer-related job positions and conduct additional experiments on it. The results can be found in “Supplementary Experiment 3: Experiments on the Designer Dataset”.

Comment 18: *How does the model perform in predicting future developments? As time is explicitly considered in the model, authors should say something on forecasting;*

Response: We are sorry for causing this misunderstanding. Actually, our model is not a forecasting model. It needs to be trained with the job postings of the current time period to obtain temporal skill embedding for skill valuation. Our model specially processes the time to assure the skills have different embedding vectors at each time interval, thus can reveal the development and change on skill semantic over time. The problem is that it causes a great increase of model complexity, which makes the model overfitting. So we use matrix factorization and a regularizer to alleviate this problem. That is why the process of time seems special. Still, it is a good idea to forecast the future trend of skill value, which can be a useful tool for predicting the future trends of labour markets. Indeed, in some of our previous studies [9, 10], we have developed methods to forecast the trend

of skill demand in the job market. Nevertheless, we think predicting the skill value is a more complicated task, which we will explore in the future works.

Sorry again for this confusion. In the revised version, we have revised the relevant descriptions. Specifically, we added the following descriptions to the “Temporal Skill Embedding” Section in the “Methods” Section:

With the temporal skill embedding, our model can distinguish the development and change on skill semantic over time and maintains low model complexity. However, it should also be noticed that our model is not a forecasting model as training data of each time period is needed to train the corresponding embedding.

Comment 19: *Table 1 left me a bit puzzled. What scale is “value”? Also, I am not clear how it makes sense that a higher level of a skill leads to a decrease in value. This requires careful explanation and consideration. It might indicate that the model does not really work because it cannot take into account complex interaction effects (e.g., some skills occurring only in the context of other skills and then, in this specific context, being less valuable) or there are suppression effects at work. In any case, if I am not mistaken in my interpretation of Table 1, then I am concerned about the highly counterintuitive finding that a higher skill level leads to lower salary and it would be important to get to the bottom of this (might be a method effect, which then relates to the question of generalizability);*

Response: Many thanks for your comments and sorry for the confusion. The scale of value is “K RMB”, which is the same as job salary. Also, we have carefully revised the experimental results and provided more analyses on the influence level in the revised version of this paper (e.g., significance test). It turns out that though generally the influences are reasonable and consistent with Figure 3, counter-intuitive results do exist. For example, while “Know” is a relatively low level of mastery, it has positive influence on skill value when describing “JavaScript”. We looked into the original data and found the imbalanced distribution of level-skill pairs in job duties could bring bias to value estimation for some cases. Specifically, while “JavaScript” mostly appear in job related to web development, the statement “Know JavaScript” usually acts as an additional requirement for some complicated and higher-paid jobs like architecture design. Therefore, the model overestimates the skill value. Indeed, this result is explainable from a market-oriented view. Specifically, the mastery level of a specific skill usually indicates the role that it plays in the job; and therefore, the skill value highly depends on the market pricing on the relevant jobs.

We appreciate your point that due to the data bias, some special cases of skills may be unavoidably underestimated/overestimated by the model, which is a common issue for data-driven solutions. But the model will still work for the general cases. A possible solution for alleviating this kind of bias is to enlarge the diversity of the recruitment market data, which is a valuable direction for our

future studies. Indeed, you have pointed out a crucial direction for measuring skill value in a more accurate and data-efficient way. In the future, we will follow your advice and try to design new models to peel the complicated skill interactions, though it is actually a very challenging task.

In the revised version, we expanded Table 1 with more levels and significance test for better validating our findings. Then we accordingly revised the analysis of the Table as following to discuss the underestimation/overestimation problem caused by imbalance data distribution:

Nevertheless, the model also learns bias for some special cases. For example, while “Know” is a relatively low level of mastery, it has positive influence on skill value when describing “JavaScript”. The reason is that while “JavaScript” mostly appear in job related to web development, the statement “Know JavaScript” usually acts as an additional requirement for some complicated and higher-paid jobs like architecture design. Therefore, the model overestimates the skill value due to the imbalanced data distribution. Indeed, this result is explainable from a market-oriented view. Specifically, the mastery level of a specific skill usually indicates the role that it plays in the job; and therefore, the skill value highly depends on the market pricing on the relevant jobs. However, as shown in Figure 3, the model will still work for the general cases. A possible solution for alleviating this kind of bias is to enlarge the diversity of the recruitment market data, which is a valuable direction for our future studies.

In addition, we have double-checked the experimental results and found that in the last submitted version, we averaged the value of skills without level words, which causes the skill value listed in Table 1 is inconsistent with other studies of this paper. In the revised version, we have updated Table 1 with new numerical results to make the analysis more rigorous. Although there are slight changes in the numbers, the conclusion stays the same with that in the last submitted version.

Comment 20: *As authors mention throughout the manuscript, context is a relevant concept. Could authors provide additional information on how they measured context?*

Response: Thank you very much for your valuable suggestion. In the revised version, we have added descriptions about how we extract context features and feed them into the model in the supplementary information (see “Supplementary Note 2: Data Preprocessing” and “Supplementary Table S4”).

Comment 21: *Would it be possible for authors to report more long-term data (e.g., from the last 10 or 20 years)? This could give interesting insights on general developments on the relevance of different skills for different markets;*

Response: Many thanks for your advice! We would like to collect long-term data to show more interesting findings with our model. Indeed, our data-driven model

is based on the accumulated job advertisements in online recruitment websites. However, the online recruitment service only has a short history, we are not able to obtain the real-world data for longer period of time. This certainly could be improved in the future. We plan to continuously update our result so that we can provide long-term analysis for the market trend.

In the revised version, we have discussed this problem in the “Limitation” Section.

Comment 22: *Many of the results are interesting, for instance the ones in Figure 3. However, I noted that many of the interpretations rely on analyzing descriptive patterns in the data by the mere looks of it. This comes along with a high chance of capitalizing on random trends and findings (albeit the large data set) and of being inaccurate. Maybe authors could choose a more formal approach throughout the manuscript to test their assumptions?*

Response: Thank you very much for your valuable comments. We fully understand your concern and we do want to validate our result in a more quantitative and formal way. However, measuring skill value from job advertisements is a challenging open problem. As we have proposed a kind of measurement that has not been explored before, we do not have a widely-accepted quantitative measurement of market-oriented skill value to quantitatively validate the value accuracy. Therefore, in this paper, on one hand, we look at the result and see if the value is reasonable; and on the other hand, we use salary prediction performance to indirectly validate the performance of skill valuation. The rationale behind is that since we have explicitly formulated the relationship between salary and skill, the effectiveness of skill value will directly influence the salary prediction performance. Despite all the reasons we cannot provide a formal approach to test the assumptions, we fully understand your concern. So we try our best to show the results from more views. In the revised version, we have added more analysis and visualizations to provide more insights into our model, which can be found in the supplementary information.

Specifically, we have added the variance in the figures (Figure 3 and the Supplementary Figure S6) to give more information on the value distribution. We have expanded an extra dataset containing designer-related job postings and have given skill value visualization on it (see “Supplementary Experiment 3: Experiments on the Designer Dataset”). To show the city-aware skill value, we have given a rank for programming skills in six different big cities (see Supplementary Table S9). We also analyzed the skill value distribution on different occupations to show our model assigns overall skill value estimation that is shared by all the occupations (see Supplementary Figure S9). Besides, we have added the numerical statistics of the figures in the supplementary information.

Comment 23: *I like the notion that many jobs require generic skills and that this type of skill set applies more broadly to different jobs as distinct and rather narrow skills. This could be made stronger in the manuscript and could be connected to the vast literature on 21st century society, educational changes in a digital world, and how the employment market has been changing over the last decades;*

Response: We very much appreciate your constructive suggestions. Following your advice, in the revised version, we have introduced more related works and provided the discussions about generic and specific skills in the paper.

For your convenience, we show the revisions as follows:

Indeed, most jobs on the market are not so professional and dominated by some generic skills. In this case, some high-value skills may also be involved, but usually not a major part of work. Also, new skills are emerging rapidly with all the fast technology changes, which enlarges the skill gap between job candidates and employers [11]. From the viewpoint of the employers, although there are few candidates who have the required specific skills, the talents owning generic skills may be able to quickly learn and adapt to the required skills [12]. Accordingly, higher education in recent years have been focusing on teaching theoretical and basic knowledge, and cultivating students' learning ability and problem-solving skills rather than teaching specific skills [13].

Comment 24: *Figure 4 provides nice illustrations but is very descriptive. Could the information contained in this figure be calculated/presented using actual statistics?*

Response: Thank you for the advice. In the revised version, we have introduced more numerical analysis, and provided all the numerical data related to the figures in the supplementary information (see “Supplementary Tables: Numerical Statistics of the Original Figures”). In addition, we provided the source data used to draw the figures, and excel files that contain the statistics of the figures as the supplementary file.

Comment 25: *Table 3: I am not clear about the scale for columns 2 and 3;*

Response: The scale is “K RMB”. We have made this clear in Table 3.

Comment 26: *The next step in the research presented by the authors would be to have a model that gives an estimation of a fair salary (both for employer and employee) and also provides guidance for applicants on where to apply given their job profile. Authors do mention this as a possibility, but I wonder whether it wouldn't be possible to present such an equation. In any case, authors should discuss the relevance of such a model and of their approach in way that it takes into account broad potential applications;*

Response: Thank you very much for the advice! In the revised version, we have added discussions about potential applications. Specifically, we have discussed three potential applications of our work, including salary estimation, job seeking guidance and job skill learning guidance for career development (in the “Potential Applications” of “Discussion” Section).

Specifically, we have added the following descriptions:

In the experiments, we show that the skill valuation model has the potential to be applied to various real-world applications. First, as can be observed in Table 2, our model achieves high performances on salary prediction. Therefore, it can provide salary reference in job the market when the job description is specified. With the predicted salary information, the recruiters can understand about the market competitiveness of their offered salaries and the job seekers can get an idea about their salary expectations. Second, as shown in Figure 3, our model achieves fine-grained skill valuation with the awareness of various contexts. Based on the skill value, problem-specific metrics can be designed for more applications with context-awareness. For example, by measuring the average value of skills in different companies, as shown in Figure 3 (d), job seekers can receive effective guidance on which company is more suitable for them to pursue. Third, the skill value also provides the students with the market-oriented guidance for skill learning. For example, with the experience-aware skill value shown in Figure 3 (c), students are able to make better personalized curriculum choice to achieve long-term career development.

Comment 27: *Case study: I have doubts that this can be applied to individual job postings. Authors should distinguish between averaged postings in the model and applying it to individual cases;*

Response: Sorry for the misunderstanding. In fact, the model can be operated on individual job postings. For each job posting, the model takes the contextual features and skill set as the input. Then, it predicts the value of each skill under the specified context and calculates the skill domination according to the skill co-appearance. Finally, it combines the skill value into the salary prediction for the jobs according to the domination. The above shows the process how individual job postings are decomposed by the model. Indeed, while we train the model parameters with a set of job postings, the model can be applied to individual job postings. Then, since the skill value is modeled independently from job postings, the obtained skill valuation model can return context-aware skill value which is shared by all the job postings. Moreover, the skill domination is job-specific since it is related to the co-appearance of other skills. Thus, the skill domination shown in the word cloud is averaged on all the job postings. In the case study, we simply fed a job posting to the model and obtained the value and job-specific domination. This is exactly how the model works. We are sorry for the confusion. In the revised version, we have refined the relevant descriptions.

In particular, we have added the following descriptions in the “Case study on a job posting” Section:

For each job posting, SSCN predicts the value of each skill under the specified context, calculates the skill domination based on the skill co-appearance, and finally combines the skill value into the salary prediction for the jobs according to the domination. Figure 4 (d) shows the case study to illustrate how SSCN works on a job posting. Specifically, we used the trained SSCN to decompose a randomly selected job posting and illustrated the domination, contribution, and overall influence of the skills on the salary.

3. Response to Reviewer 2

Comment 1: *This paper addresses the problem of evaluating and predicting the value of a skill as it emerges by processing and analysing online job postings (aka, online job vacancies/advertisements). The problem is relevant and has a practical significance for the industry, the academy, and for the governments, too. The paper is well written and contextualised. However, I see two main issues related to (i) significance of the results and (ii) technical Contribution (see comments below).*

Response: We very much appreciate your recognition on this paper, especially for its broad significance. In the following, we will introduce the details of our revision and address your main concerns through the point-by-point response.

Comment 2: *The authors seem to do not be familiar with the domain of Labour Market Intelligence, that employs AI algorithms to derive knowledge useful for understanding labour market dynamics and trends.*

Response: Thank you very much for your suggestion and sorry for missing related works in the last submitted version. According to your advice, we have surveyed the related works on Labour Market Intelligence (LMI). Although not directly working on the field of LMI, the major authors of this paper have been working on data-driven talent analysis and recruitment-related analysis (e.g., Talent and Management Computing [21]) for many years, such as recruitment market demand analysis [9,10,14], talent flow analysis [15, 16], company profiling [17], salary prediction [7], job title benchmarking [18], etc. Some of these works are very relevant to the topics of LMI. According to your comments, in the revised version, we have added a new section to describe the related works of this paper, including LMI and other relevant topics, which can be found in the supplementary information (see “Data-driven Labour Market Analysis” in “Supplementary Note 1: Related Works”).

Comment 3: *They do not discuss the relevance and impact of processing vacancies for understanding labour market dynamics indeed (see, e.g.: Frey, C.B., Osborne, M.A.: The future of employment: How susceptible are jobs to computerisation? Technological Forecasting and Social Change 114(Supplement C), 254 – 280 (2017) ; Alabdulkareem, A., Frank, M.R., Sun, L., AlShebli, B., Hidalgo, C., Rahwan, I.: Unpacking the polarisation of workplace skills. Science advances 4(7) (2018)) as well as the importance for governments given last relevant projects on collecting and processing job posting skills (see, e.g. <https://www.cedefop.europa.eu/it/about-cedefop/public-procurement/real-time-labour-market-information-skill-requirements-setting-eu>; https://ec.europa.eu/eurostat/cros/content/essnet-big-data_en;)*

Response: Many thanks for your valuable suggestions. Following your advice, in the revised version, we have revised the descriptions about the relevance of our work with understanding labour market dynamics and its significance for governments. Besides, we have introduced more related works in the supplementary information.

Specifically, we have revised the first paragraph of the introduction as following descriptions and have given citations about related works in the manuscript:

At the micro level, it can not only help individuals to proactively assess their competencies and decide what are the right skills to learn, but also help companies to develop the right salary system of their job positions for attracting and retaining the best possible talent. Moreover, at the macro level, the job skill value is an important indicator of the economic equilibrium of labour market and shows the supply and demand relationship associated with knowledge investments.

And we have described the related work of three parts, including data-driven labour market analysis, data-driven salary prediction, and attentive neural networks. In “data-driven labour market analysis” Section of “Supplementary Note 1: Related Works”, we have described job advertisement data as “*provides an unparalleled chance for catching the labour market dynamics in an automatic and cost-efficient manner*” and cited some related works.

Comment 4: *Furthermore, some recent and relevant works are omitted. Specifically, the authors missed comparing with a very similar approach that employs vector-space models for salary calibration-prediction (Zhang, D., Liu, J., Zhu, H., Liu, Y., Wang, L., Wang, P., Xiong, H.: Job2vec: Job title benchmarking with collective multi-view representation learning. In: CIKM. pp. 2763–2771 (2019)).*

Response: We very much appreciate your suggestion and sorry for the confusion. The mentioned paper (i.e., Job2vec: Job title benchmarking with collective multi-view representation learning) aims to use talent flow data for job title benchmarking instead of salary benchmarking, which cannot be used as a baseline for the salary prediction task (actually, the corresponding authors of the

work you have mentioned are also the main authors of this paper). We have surveyed the problem of salary prediction. Although there are related works to predict the salary, little of them fit for the application scenario of this paper that aims to predict the salary for job postings. Therefore, we apply machine learning techniques that are widely-adopted in real-world prediction tasks (i.e., SVM, LR, GBDT), the most recent salary benchmarking method HSBMF [7] published in ICDM'2018 (i.e., one of the premium data mining conferences), and variants of our proposed model as comparisons in the experiment. The experimental result demonstrated that our model outperforms these methods.

According to your comment, in the revised version, we have added a supplementary section for introducing the related works of salary prediction (see “Data-driven Salary Prediction” in “Supplementary Note 1: Related Works”).

Comment 5: *The authors do not discuss at all *how* information that they use in their model are extracted from job postings (aka, online job advertisements). They just mention the online source from which they have been collected, without providing any information about the quality (update time, missing values, num of duplicated vacancies, coverage over all occupation types, etc.)*

Response: Many thanks for your comments and sorry for missing this information. In the revised version, we have added descriptions about data preprocessing (see “Supplementary Note2: Data Preprocessing”) and data analysis (see “Supplementary Experiment 1: Data Analysis”). Specifically, Supplementary Figure S1, Supplementary Figure S2 and Supplementary Table S6 show the city distribution of our job posting data. Figure S3 shows the salary distribution. Supplementary Figure S4 shows job title distribution. Supplementary Table S5 shows statistics of the two datasets.

Comment 6: *Indeed, the problem of extracting such information such as (i) company name (ii) location, (iii) salary is far from straightforward, and it mainly affects the trustability of the analyses based on that information.*

Response: Many thanks for your valuable comments and sorry for the confusion. Actually, the postings crawled from the mentioned recruitment website is naturally structured data in the form of html. By parsing the html, information like company name, location, and salary can be directly obtained. We very much thank you for reminding us to describe it. In the revised version, we have added descriptions of the raw data and related information extraction in the supplementary information (see “Supplementary Note 2: Data Preprocessing”).

Comment 7: *Authors say they trained their model on a "large-scale" set of job postings composed by 800k vacancies over 36 months. However, those publication*

rate is shallow compared with the publication rate of the major EU websites (see <https://www.cedefop.europa.eu/en/data-visualisations/skills-online-vacancies>) and this limits both the effectiveness of your approach and the significance of the results (intended as the value of skills they aim to assess).

Response: Thank you very much for your valuable suggestions. Indeed, we fully believe that, with more data sources, the skill valuation will be more accurate. However, this research has certain requirements of data quality. Specifically, the job postings should involve abundant skill requirements and have salary and detailed job contextual information. This raises the difficulty of data collection. As a result, we only make it to collect a representative Chinese IT job advertisement dataset. Although it is true that limited data will bring bias to the model, since our dataset covers the Internet-related industry of China, we believe it can support a research work of skill valuation model. Through the experiment, the effectiveness of our model has been validated.

To the best of our knowledge, the data we have used is the largest open-access data that satisfy our requirements. It is larger than similar datasets used in works published in the top data mining conferences [9, 15, 19, 20]. In particular, we guarantee to publish this dataset once the paper is accepted. Meanwhile, we are sorry for not familiar with the data source of the website that you mentioned. Actually, we have investigated this website, while unfortunately did not find open-access data for conducting our experiments. Nevertheless, we have to admit that compared with government websites that conduct statistical analysis on data collected from various data sources, the scale of our dataset is still quite limited. In the future, we will keep collecting more data to improve our analysis.

In the revised version, according to your comments, we have deleted the "large-scale" description and added discussions about the limitation of the data scale (see "Limitation" Section). To better demonstrate the effectiveness and generalization of the proposed model, we also collected an extra dataset and conducted additional experiments, as shown in the supplementary information (see "Supplementary Experiment 3: Experiment on The Designer Dataset"). We hope that our model can be applied to larger datasets in the future.

Comment 8: *Furthermore, they do not discuss in any way.*

(i) the type of online source, if you include aggregators, the number of duplicate vacancies grows a lot (up to 30% according to a recent study), how did you identify duplicate vacancies?

Response: Many thanks for your comments and sorry for missing the descriptions of the data source. In the revised version, we have added this information to the supplementary information. The dataset is not collected from an aggregator but directly from the online recruitment platform. Following your advice, we have added descriptions of identifying the duplicated job postings in the supplementary information (see "Duplication Processing" in "Supplementary

Note 2: Data Preprocessing”). In short, since our data are structured, we can compare the contextual information and job salary of job postings to each other. Then we further compare the edit distance and semantic similarity of the job descriptions to recognize the duplicated job postings. Actually, there are little duplicated vacancies (less than 1%) in our data. For better understanding, in the revised version, we have shown this information in Supplementary Table S3.

Comment 9: *empirically, the salary information (that is crucial in your model) is present in only 15-20% of vacancies. Which is the distribution of salary variable within vacancies? how has this information been collected?*

Response: Sorry for the confusion due to the lack of corresponding information. Since the website where we collected the data requires the recruiter to offer a salary range when posting a job advertisement, there is little missing salary information. Indeed, enough salary information is extremely crucial for the data to support our study.

During our experiment, we also tried to collect data from recruiting websites in other countries. However, we found the same phenomenon as you have mentioned. It is true that the salary information is optional in many online recruitment websites, so there are only 15-20% of vacancies that have salary information.

We very much thank you for your constructive advice. In the revised version, we have given more detailed descriptions of the raw data (see “Data Source” in “Supplementary Note 2: Data Preprocessing”). Then we have given some visualizations of analysis (with salary distribution in Supplementary Figure S3) and a table of statistics of the raw data (Supplementary Table S5).

Comment 10: *For comparing and analysing vacancies, the most important step is to classify vacancies over standard occupation taxonomies (e.g., ISCO/ESCO for Europe, O*Net for US, see e.g. Boselli, Cesarini, Mercorio, Mezzanzanica: Classifying online Job Advertisements through Machine Learning. Future Gener. Comput. Syst. 86: 319-328 (2018)). This is a crucial to understand if analyses (here, the Salary-Skill Value Composition analysis) is uniformly distributed over sectors/occupations, as well as to assess if your ads set is overestimating/underestimating some occupations/sectors. Otherwise, the analyses represent a technical exercise that does not give significant benefit to the user as it cannot explain the reality.*

Response: Many thanks for your suggestions. In this paper, we focus on assessing the value of skills for the overall job market, which are shared by all the occupations. If a more fine-grained analysis is required, the occupation taxonomy can be regarded as a context in our model. Then the model can provide occupation-aware skill value. Since our dataset does not naturally contain the

field of occupation and the job title are customized by the recruiters, it is a non-trivial task to classify them into the standard occupations. To avoid error accumulating in the experiment, we did not do that in this paper. Following your advice, in the revised version, we have analyzed the job titles of our dataset (see Supplementary Figure S4), which can reveal the distribution of occupations. Besides, you have inspired us to analyze the value distribution of the current model's assessment among different occupations. We manually grouped some frequently appeared job titles of some occupations and observed that the value distributes similar on different occupations. This proves our model to be assigning overall skill value for the market. This result can be found in Supplementary Figure S9.

Comment 11: *The problem definition is intuitive, and the article is undoubtedly well written. However, I am ambivalent about the strength of its theoretical Contribution. The proposed method for learning skill value is a straightforward application of existing work. The Contribution is thus limited to the use of such techniques in a new domain and task. Even though the use of attention and cooperative networks in Labour Market is quite new, this is an application paper without major technical novelty.*

Response: Many thanks for your comments and sorry for the misunderstanding. In this work, we focus on the skill valuation problem and propose a novel framework. To our best knowledge, it is the first work to value skills from the perspective of modeling their specific influence on salary. The main challenge of this problem is the lack of ground truth for training neural network models. So our major technical contribution is the data-driven framework to utilize indirect supervision (salary) for the targeted model training (skill valuation). Then we specifically designed a network structure to achieve this transfer of supervision, which is a novel trial and we think it is a technical contribution. We have to admit that compared with research works of machine-learning theories, this paper has limited theoretical contributions. However, we believe our problem formulation and insights from the results are appealing to the audiences. In the revised version, we have enriched corresponding descriptions to further clarify the technical contribution and limitation of this paper. Specifically, in “Salary-Skill Value Composition Problem” Section, we have revised the following descriptions to better understand the technical challenge of this work:

To train the model, it is essential to obtain a set of training data containing job postings that only require one skill. However, in the real-world scenario, the job requirements are always complicated and cannot be qualified with only one skill. As a result, each job posting is always associated with multiple required skills, which makes it difficult to train the skill valuation model under supervised learning paradigm.

Then, we revised Figure 1, together with the caption, to better convey the technical contribution of our model.

Comment 12: *Experiments are not robust. Hyperparameter tuning for the author's model and baseline models is missing,*

Response: Thank you very much for your constructive suggestions. In the revised version, we have added the results of the hyperparameter experiment in the supplementary information (see “Supplementary Experiment 2: Parameter Experiments”). It turns out that our model is robust and not sensitive to the hyperparameter. As for the baselines, since most of the traditional regression models are not sensitive to parameters, we simply used the commonly adopted default hyperparameters.

Comment 13: *and so validation strategy (cross-validation, holdout, how overfitting is missing, etc.),*

Response: Sorry for the confusion due to our unclear description. Actually, we did 10 times of holdout validation for salary prediction and showed the mean and standard deviation of the metrics in Table 2. We are sorry that in the last submitted version, we only roughly described this process in the caption, which obviously has confused the readers. Following your advice, we described the validation more in detail in the revised version, which can be found in “Methods” Section.

Specifically, we have added “Validation” in “Methods” Section with the following descriptions:

We repeated 10 times of hold-out validation on the models. Specifically, at each time, we randomly split the data into training and testing set with a ratio of 4:1. We use the training data to train the models and use the testing data to evaluate the model performance.

Comment 14: *weights initialisation, etc. Those are usually essential details when dealing with neural networks, especially the complex model presented by the authors.*

Response: Many thanks for your suggestion. The weights are initialized with glorot normal initializer. In the revised version, we have added descriptions of the weight initialization in “Network Configuration” of the “Methods” Section.

Comment 15: *Related works about collaborative neural networks and attention networks are missing.*

Response: Many thanks for your comments. Following your advice, in the revised version, we have added survey on related works about attentive networks in supplementary information. As for the collaborative neural network, we surveyed it and found little related papers. Do you mean the term “cooperative neural network” we used in our paper? If so, we call our network as “cooperative” since it achieves the major unsupervised task of skill value with the cooperation of the salary prediction task. Following your advice, in the revised version, we have introduced more related works in “Supplementary Note 1: Related Works”.

Comment 16: *Minors.*

Response: Many thanks for pointing out these problems for us. In the revised version, we have carefully polished the use of language by correcting typos, grammar errors and confusing expressions.

References

- [1] Dix-Carneiro, R., & Kovak, B. K. (2015). Trade liberalization and the skill premium: A local labor markets approach. *American Economic Review*, 105(5), 551-57.
- [2] Burstein, A., & Vogel, J. (2017). International trade, technology, and the skill premium. *Journal of Political Economy*, 125(5), 1356-1412.
- [3] Topel, R. H. (1994). Regional labor markets and the determinants of wage inequality. *The American Economic Review*, 84(2), 17-22.
- [4] Han, J., Pei, J., & Kamber, M. (2011). *Data mining: concepts and techniques*. Elsevier.
- [5] Dean, J. (2014). *Big data, data mining, and machine learning: value creation for business leaders and practitioners*. John Wiley & Sons.
- [6] Bose, I., & Mahapatra, R. K. (2001). Business data mining—a machine learning perspective. *Information & management*, 39(3), 211-225.
- [7] Meng, Q., Zhu, H., Xiao, K., & Xiong, H. (2018, November). Intelligent salary benchmarking for talent recruitment: A holistic matrix factorization approach. In *2018 IEEE International Conference on Data Mining (ICDM)* (pp. 337-346). IEEE.
- [8] Blankmeyer, E., LeSage, J. P., Stutzman, J. R., Knox, K. J., & Pace, R. K. (2011). Peer-group dependence in salary benchmarking: a statistical model. *Managerial and Decision Economics*, 32(2), 91-104.
- [9] Zhu, C., Zhu, H., Xiong, H., Ding, P., & Xie, F. (2016, August). Recruitment market trend analysis with sequential latent variable models. In *Proceedings of the 22nd ACM SIGKDD international conference on knowledge discovery and data mining* (pp. 383-392).
- [10] Xu, T., Zhu, H., Zhu, C., Li, P., & Xiong, H. (2017). Measuring the popularity of job skills in recruitment market: A multi-criteria approach. *arXiv preprint arXiv:1712.03087*.
- [11] McGowan, M. A., & Andrews, D. (2015). Skill mismatch and public policy in OECD countries.
- [12] Badcock, P. B., Pattison, P. E., & Harris, K. L. (2010). Developing generic skills through university study: a study of arts, science and engineering in Australia. *Higher education*, 60(4), 441-458.
- [13] National Research Council. (2012). *Education for life and work: Developing transferable knowledge and skills in the 21st century*. National Academies Press.
- [14] Wu, X., Xu, T., Zhu, H., Zhang, L., Chen, E., & Xiong, H. (2019, August). Trend-Aware Tensor Factorization for Job Skill Demand Analysis. In *IJCAI* (pp. 3891-3897).

- [15] Xu, H., Yu, Z., Yang, J., Xiong, H., & Zhu, H. (2018). Dynamic talent flow analysis with deep sequence prediction modeling. *IEEE Transactions on Knowledge and Data Engineering*, 31(10), 1926-1939.
- [16] Zhang, L., Xu, T., Zhu, H., Qin, C., Meng, Q., Xiong, H., & Chen, E. (2020, April). Large-Scale Talent Flow Embedding for Company Competitive Analysis. In *Proceedings of The Web Conference 2020* (pp. 2354-2364).
- [17] Lin, H., Zhu, H., Zuo, Y., Zhu, C., Wu, J., & Xiong, H. (2017). Collaborative company profiling: Insights from an employee's perspective. *arXiv preprint arXiv:1712.02987*.
- [18] Zhang, D., Liu, J., Zhu, H., Liu, Y., Wang, L., Wang, P., & Xiong, H. (2019, November). Job2Vec: Job title benchmarking with collective multi-view representation learning. In *Proceedings of the 28th ACM International Conference on Information and Knowledge Management* (pp. 2763-2771).
- [19] Yan, R., Le, R., Song, Y., Zhang, T., Zhang, X., & Zhao, D. (2019, July). Interview choice reveals your preference on the market: To improve job-resume matching through profiling memories. In *Proceedings of the 25th ACM SIGKDD International Conference on Knowledge Discovery & Data Mining* (pp. 914-922).
- [20] Liu, L., Liu, J., Zhang, W., Chi, Z., Shi, W., & Huang, Y. (2020, July). Hiring Now: A Skill-Aware Multi-Attention Model for Job Posting Generation. In *Proceedings of the 58th Annual Meeting of the Association for Computational Linguistics* (pp. 3096-3104).
- [21] 2020 International Workshop on Talent and Management Computing. Held in conjunction with KDD'20. August 24, 2020 - San Diego, CA, USA. <http://bigdata.usc.edu.cn/TMC2020/>

Reviewers' Comments:

Reviewer #1:

Remarks to the Author:

Thank you for giving me the opportunity to review the revised version of the manuscript entitled "Market-oriented Job Skill Valuation with Cooperative Composition Neural Network." In their work, authors aimed at introducing a new way of value assessment of job skills. More specifically, authors analyzed job adverts from one large recruitment site and the profiles defined therein. They subsequently used machine learning algorithms to come up with a model that can predict the value of different job skills defined as their direct effect on salary when taking the context of a position into account.

I was one of the two initial reviewers and have now carefully looked at authors' revisions both in terms of the rebuttal letter and the revised manuscript. I would again like to disclose that I am no expert in the methods that the authors apply so please consider this when reading my methodological comments. I hope that the other referee can give competent feedback on the specific methods (and the revisions along these lines) that were employed by the authors.

My impression is that the manuscript has greatly improved throughout the revision process and that it is moving towards publication. I felt that the answers to my comments were reasonable and that adequate changes have been implemented by the authors. There are a couple of points that authors should integrate when further revising the manuscript but they are of a rather minor nature:

Comment 1:

There are still some oddities and minor language glitches. Please make sure that those are fixed.

Comment 2:

I am referring to this previous comment:

Comment 19: Table 1 left me a bit puzzled. What scale is "value"? Also, I am not clear how it makes sense that a higher level of a skill leads to a decrease in value. This requires careful explanation and consideration. It might indicate that the model does not really work because it cannot take into account complex interaction effects (e.g., some skills occurring only in the context of other skills and then, in this specific context, being less valuable) or there are suppression effects at work. In any case, if I am not mistaken in my interpretation of Table 1, then I am concerned about the highly counterintuitive finding that a higher skill level leads to lower salary and it would be important to get to the bottom of this (might be a method effect, which then relates to the question of generalizability);

I acknowledge authors' response and it clarified my original comment to some extent. However, I think some quantification is needed as to which percentage of job profiles might be affected by a high level of error (due to potential interaction effects). The authors mentioned that in the original version of the manuscript, one such example was present, which I flagged in my original review (and which led to comment 19). Apparently, authors consider this an exception but how can they be sure about this? Please provide some quantification, justification, and some compelling data that for most of the job profiles your model works.

Comment 3:

In Table 1, it reads "significant" and "very significant". These terms are not used in statistical reporting. Just reporting the p-values should do the job here.

Comment 4:

I acknowledge that the authors extended the potential applications of their research in the final

paragraph of the main body of the manuscript, but this part needs to be strengthened further. In their rebuttal letter, authors give some examples (and I agree with them), but not all of them have been convincingly put into the revised manuscript.

Apart from this, I was happy with the revisions that authors made and I wish them best of luck with continuing their research.

Reviewer #2:

Remarks to the Author:

The contribution of this work, from what I can understand and it has made cleared by the authors in the rebuttal, is to train a supervised model using job salary as a target value, in order to predict the salary for job posts, and then decompose this value in the required skills.

The approach is really interesting and more sound after this review and encompasses several interesting insight (e.g. impact of experience and company on job salary). However, a few issues still puzzle me:

- The contribution, as confirmed by the authors in the rebuttal, is quite limited, even though the method is sound and well developed and the insights and results can be of interest for a broader audience (they are to me!). I think it should be the editor and the committee to judge if this research fulfils the requirement of the journal in terms of novelty and impact (800k job ads, of which 200k selected from a single source over 4 years. This means the results retrieved have a very limited significance and cannot in any way represent the salary dynamics of your Country.) Given the audience of the journal and the target of the paper, I think this is the main weakness of this study.

- As highlighted by the authors, the literature regarding the job salary prediction task is scarce. Nonetheless, I believe the authors should compare their model on this task with the state of the art models regression in text mining. This would be a fair comparison because, all in all, the task of job salary prediction is a regression task.

- Also for the skill salary decompositions, I understand the literature is inadequate, but still, the problem of feature decomposition can be analysed from many perspectives, with feature relevance/XAI methods. Which is the contribution of your approach? Is it comparable with others? It would be interesting to find a way to evaluate also the skill decomposition task.

Overall, I have the feeling that the paper, after the rebuttal, is still kind of self-contained and is difficult to assess its impact both to the Labour Market domain (regarding the job value prediction task) and also regarding the problem of feature domination assessment (skill domination task).

Summary of Revisions

Title: Market-oriented Job Skill Valuation with Cooperative Composition
Neural Network

Submission ID: NCOMMS-20-27613d

Authors: Ying Sun, Fuzhen Zhuang, Hengshu Zhu, Qi Zhang, Qing He, and
Hui Xiong

First of all, we would like to thank the editor and two anonymous reviewers for providing us an opportunity to further improve this paper. Your constructive comments have not only helped a lot on the improvement of this paper in a number of ways, but also given a lot of inspirations for our future study. In the following, we first summarize the major revisions of this paper, and then address the specific comments from the two reviewers. As a convention, the “last submitted version” refers to the version we submitted in November 2020 for the second-round review, and the “revised version” refers to the current submitted version.

1. Revision Summary

We have revised the paper according to the comments from the editor and two reviewers. The revised details have been summarized as follows.

In the revised version, we have provided substantial new content:

- We have provided more analysis for better supporting our demonstrations and added more supplementary information.
- We have further discussed the potential applications of this work.
- We have conducted experiments with state-of-the-art text-based baselines to further validate the effectiveness and advantages of the proposed model.

In the revised version, we have revised the original content:

- We have further highlighted the technical contributions of this work.

- We have revised some figures and tables (e.g., Table 1 and Supplementary Figure S5), which are consistent with the ones in the last submitted version but are more informative.
- We have revised some descriptions, formats, and texts for improving the readability of this paper. Also, we carefully polished the use of language. For example, we corrected typos, grammar errors, and confusing expressions in the revised version.

2. Response to Editor

Comment: *Please note that, while we disagree with Reviewer 2 on the impact and novelty of your work, we would like to see all technical comments addressed, including benchmark with a standard text mining regression method, as suggested by the Reviewer. We are convinced that ensuring methodological novelty is essential here to prove the interest to a broad community.*

Response: We very much appreciate your recognition on the significance of this work. In this paper, we have studied a cross-disciplinary social-economic problem; that is, how to assess the value of job skills with data-driven machine learning techniques from a market-oriented perspective. From the technical view, we introduce a novel cooperative framework to train neural network models for unsupervised learning tasks, by quantitatively linking them with a supervised learning task. This cooperative framework can potentially be used for broad application domains with similar learning scenarios, such as business market analysis, talent recruitment and education.

In the revised version, we have addressed all the technical comments. Specifically, we have further clarified our technical contribution (in the “Introduction” Section and the “Discussion” Section) and compared our model with state-of-the-art text-based regression methods (Table 2 and Supplementary Table S8) according to the suggestions of the reviewers. Meanwhile, we have updated Supplementary Figure S5 according to the added baselines.

3. Response to Reviewer 1

Comment 1: *My impression is that the manuscript has greatly improved throughout the revision process and that it is moving towards publication. I felt that the answers to my comments were reasonable and that adequate changes have been implemented by the authors. There are a couple of points that authors should integrate when further revising the manuscript but they are of a rather minor nature:*

Response: Thank you very much for your positive feedback on this paper! According to your suggestions, we have carefully revised this paper. In the following, we will introduce the details of our revision and address your remaining concerns through the point-by-point response.

Comment 2: *There are still some oddities and minor language glitches. Please make sure that those are fixed.*

Response: Many thanks for your suggestions! In the revised version, we have further polished the use of language by correcting typos, grammar errors, and confusing expressions. If necessary, we are also willing to seek help from the language editing services to improve our presentations before publication.

Comment 3: *I am referring to this previous comment:*

Comment 19: Table 1 left me a bit puzzled. What scale is “value”? Also, I am not clear how it makes sense that a higher level of a skill leads to a decrease in value. This requires careful explanation and consideration. It might indicate that the model does not really work because it cannot take into account complex interaction effects (e.g., some skills occurring only in the context of other skills and then, in this specific context, being less valuable) or there are suppression effects at work. In any case, if I am not mistaken in my interpretation of Table 1, then I am concerned about the highly counterintuitive finding that a higher skill level leads to lower salary and it would be important to get to the bottom of this (might be a method effect, which then relates to the question of generalizability);

I acknowledge authors’ response and it clarified my original comment to some extent.

However, I think some quantification is needed as to which percentage of job profiles might be affected by a high level of error (due to potential interaction effects). The authors mentioned that in the original version of the manuscript, one such example was present, which I flagged in my original review (and which led to comment 19). Apparently, authors consider this an exception but how can they be sure about this? Please provide some quantification, justification, and some compelling data that for most of the job profiles your model works.

Response: Many thanks for your valuable suggestions! In the revised version, we have rigorously analyzed this problem and proposed a method to precisely calculate the ratio of exception for all the skills in our dataset. Specifically, given the partial ordering sets of general and skill-specific level influence, we designed an algorithm to find out the skill-level observations that have caused the inconsistency between the two sets. As a result, only 0.96% of the samples in our dataset are exceptions, which further supports our previous conclusions.

According to your suggestions, in the revised paper, we have added the following descriptions:

Furthermore, we calculated the ratio of skill-level observations that might cause the biased level influence estimations. The result shows that only very few samples (0.96% of the whole dataset) encounter this bias. The detailed calculation can be found in the Supplementary Information.

Meanwhile, we have added detailed descriptions in the Supplementary Information (see “Supplementary Note 3: Level Influence Bias Estimation”) to introduce the details of the estimation, including the problem formulation and the pseudo-code of our calculation algorithm.

Comment 4: *In Table 1, it reads “significant” and “very significant”. These terms are not used in statistical reporting. Just reporting the p-values should do the job here.*

Response: Thank you very much for the suggestion and sorry for the confusion. In the revised version, we have reported the p-value of each entry in Table 1.

Comment 5: *I acknowledge that the authors extended the potential applications of their research in the final paragraph of the main body of the manuscript, but this part needs to be strengthened further. In their rebuttal letter, authors give some examples (and I agree with them), but not all of them have been convincingly put into the revised manuscript.*

Response: Many thanks for your suggestion! According to your suggestion, in the revised version, we have added more discussions on the potential applications of our work. Specifically, we have discussed the potential impact of our method on talent recruitment, business market analysis, student education, knowledge management, and job recommendation.

Specifically, we have revised the “Potential applications” Section as follows:

Through the experiments, we show that the skill valuation model has the potential to be applied to various real-world applications. First, our model can be applied to talent recruitment. As can be observed in Table 2, our model achieves high performance on salary prediction. Therefore, it can provide salary references for jobs in the market when the job descriptions are specified. With the predicted salary information, the recruiters can evaluate the market competitiveness of their offered salaries; and the job seekers can get an idea about their salary expectations. Second, our model can be directly applied to business market analysis. For example, as can be observed in Figure 3 (b), our model reveals the overall trend of skill value in the market. Third, our model can be applied to student education. Specifically, the skill

value provides the students with market-oriented guidance for skill learning. For example, with the experience-aware skill value shown in Figure 3 (c), students are able to make better personalized curriculum choices to achieve long-term career development. Fourth, our model can be applied to knowledge management and talent development. For example, as shown in Figure 3 (d), the companies can analyze the value of skills for their own business. Then, they can develop specific curriculums to continuously train their employees for valuable skills. Fifth, our model can be applied to job recommendation. For example, by measuring the average value of skills in different companies, as shown in Figure 3 (d), job seekers can receive effective guidance on which company is more suitable for them to pursue.

Comment 6: *Apart from this, I was happy with the revisions that authors made and I wish them best of luck with continuing their research.*

Response: Many thanks for all of your professional suggestions, which indeed have helped a lot on improving this paper!

4. Response to Reviewer 2

Comment 1: *The contribution of this work, from what I can understand and it has made cleared by the authors in the rebuttal, is to train a supervised model using job salary as a target value, in order to predict the salary for job posts, and then decompose this value in the required skills.*

Response: Many thanks for your valuable comments and sorry for the confusion. Please allow us to further clarify the technical contribution of this work. In this paper, we have proposed a framework to jointly learn the skill valuation and salary prediction tasks. Specifically, in our approach, skill valuation is an essential part of salary prediction (i.e., the skill values are the input of the salary prediction task), while salary prediction provides the indirect supervision signal for skill valuation. Therefore, instead of decomposing the predicted salary into skill values after the salary prediction model is trained, our approach simultaneously trains neural network models for the two tasks with job posting data. In this way, we can obtain accurate job context-aware skill value with an explicit physical meaning.

Comment 2: *The approach is really interesting and more sound after this review and encompasses several interesting insight (e.g. impact of experience and company on job salary). However, a few issues still puzzle me:*

Response: We very much appreciate your recognition of this paper. In the following, we will introduce the details of our revision and address your concerns through the point-by-point response.

Comment 3: *The contribution, as confirmed by the authors in the rebuttal, is quite limited, even though the method is sound and well developed and the insights and results can be of interest for a broader audience (they are to me!). I think it should be the editor and the committee to judge if this research fulfils the requirement of the journal in terms of novelty and impact (800k job ads, of which 200k selected from a single source over 4 years. This means the results retrieved have a very limited significance and cannot in any way represent the salary dynamics of your Country.) Given the audience of the journal and the target of the paper, I think this is the main weakness of this study.*

Response: Many thanks for your valuable comments! Here, we would like to further clarify the contributions of this paper. Specifically, in this paper, we have studied a cross-disciplinary social-economic problem; that is, how to assess the value of job skills with data-driven machine learning techniques from a market-oriented perspective. As we have discussed in the manuscript, our model can potentially benefit a wide range of applications (e.g., talent recruitment, business market analysis, student education, etc.). Through our experiments, we have introduced some significant insights (e.g., how generality influences skill value and domination, how working experiences influence skill value, etc.). Actually, these insights can provide guidance for both employers and job seekers from many perspectives. Meanwhile, from the technical view, our model uses the job salary data as indirect supervision signals when the ground truth of skill value is unavailable. Indeed, the unsupervised learning tasks widely exist in many fields, where the models are expected to discover hidden patterns from the unlabeled data. Therefore, our approach provides a universal methodology for knowledge discovery with indirect supervision; that is, train neural network models for unsupervised learning tasks, by quantitatively linking them with a supervised learning task.

Indeed, since we have specific requirements on the data quality (e.g., salary information), it is costly to collect data that can be used for training our model. Therefore, compared with government-supported analytical websites, the scale of our dataset is quite limited (as we have discussed in the “Limitation” Section). However, to the best of our knowledge, the data used in this paper is the largest open-access data that satisfy the requirements of our model. Meanwhile, we promise to make this dataset publicly available for research purpose once this paper is accepted. To prove the soundness and generality of our model, we have conducted many empirical experiments, such as significance tests and experiments on two fields of occupations. These experiments show that our model can achieve significant performance and insights obtain based on our I dataset. It is reasonable to believe that, with a larger data source, our proposed model can support more accurate, fine-grained, and complicated analysis. In the future, we plan to continuously collect data and update our result so that we can provide long-term analysis for the market trend.

According to your suggestions, in the revised version, we have further clarified the contribution of this paper. Specifically, we have expanded the discussions about the potential applications of this work (see the “Potential Applications” Section). Meanwhile, we added the following descriptions in the “Introduction” Section to emphasize the general technical contributions of this work:

Indeed, SSCN provides a novel cooperative framework to train neural network models for knowledge discovery from unlabeled data, by quantitatively linking them with a supervised learning task.

Finally, we have added a “Technical Contribution” Section to further emphasize the general technical contributions of this work as follows:

Since indirect supervision is common in the real-world, we believe that this work not only provides an intelligent and accurate solution for the skill valuation problem but also can be an inspiration for readers who work on data analysis in other fields of applications. Specifically, in many real-world scenarios, obtaining labeled training data is far from an easy job. It is often the case that we can only obtain indirect supervision from a related task. Learning skill valuation model from job salary data is one of these kinds of problems. In this problem, we have no labeled data of skill value, but we have job salary data as indirect supervision information, with the intuition that high skill value usually leads to high job salary. To this end, we proposed a machine learning-based solution that uses neural network with cooperative structure to model the relationship between job and skills, where the salary prediction is regarded as a cooperative task for training the skill valuation model. In this way, we obtain an effective skill valuation model under the indirect supervision of job salary data.

Comment 4: *As highlighted by the authors, the literature regarding the job salary prediction task is scarce. Nonetheless, I believe the authors should compare their model on this task with the state of the art models regression in text mining. This would be a fair comparison because, all in all, the task of job salary prediction is a regression task.*

Response: Thank you very much for the valuable suggestions! We fully agree with you that predicting the salary with job posting texts may also be a practicable solution. Following your suggestions, in the revised version, we have compared six state-of-the-art text-based regression methods. Specifically, based on the recent studies of top conferences for Natural Language Processing (NLP) and Machine Learning (e.g., ACL, EMNLP, NeurIPS, etc.), we have designed two groups of baselines. The first group consists of well-adopted NLP neural network architectures trained in an end-to-end manner with our data, including:

- a. Convolutional Neural Network (textCNN) [1], which is a classic model to process textual data for information retrieval.

- b. Hierarchical Attention Network (HAN) [2], which is popular and shown to be effective for modeling long texts.
- c. Transformer-XL [3], which is a newly proposed architecture and has been widely-adopted by pre-trained models.

The second group consists of popular and state-of-the-art pre-trained models, including:

- a. Bidirectional Encoder Representations from Transformers (BERT) [5], which is the most popular pre-trained language model in recent years.
- b. Robustly optimized BERT approach (RoBERTa) [6], which is an improved model of BERT.
- c. XLNet [4], which is Transformer-XL-based pre-trained language model that has a different architecture from BERT.

In the revised version, we have added experimental results in Table 2 and Supplementary Table S8. Meanwhile, we have updated Supplementary Figure S5 according to these added baselines. Although some of these models can achieve better performance than the previous baselines, our model still outperforms them with a significant margin. This further proves the effectiveness of our model on salary prediction.

Again, thank you very much for the valuable suggestions that help us to further validate the technical soundness of this paper.

Comment 5: *Also for the skill salary decompositions, I understand the literature is inadequate, but still, the problem of feature decomposition can be analysed from many perspectives, with feature relevance/XAI methods. Which is the contribution of your approach? Is it comparable with others? It would be interesting to find a way to evaluate also the skill decomposition task.*

Response: Many thanks for your valuable comments and sorry for the confusion. Here, we would like to clarify that our approach is not a feature decomposition method, which has a different process and aims to solve a different problem from the technical view. Therefore, they are not comparable to each other.

Firstly, they have different processes. Specifically, feature relevance/XAI methods work on trained machine learning models and measure how the input features are relevant to the output with specific metrics. For example, the SHapley Additive exPlanations (SHAP) [7] regards the shapely value of features as their contribution to the output, and the Layer-wise Relevance BackPropagation (LRP) [8] designs a metric of the score backpropagated to the input layer-by-layer. Different from these works, our approach regards skill value as a component of salary prediction. Instead of decomposing the importance of skills from a trained salary prediction model, our approach jointly models skill value and its relationship with job salary. During the training process, the skill valuation model is simultaneously trained together with the

salary prediction model. This assures the skill value to be obtained by learning from job posting data instead of manually-designed metrics. Therefore, the process of skill valuation in our approach is achieved in a totally different manner from the feature relevance/XAI methods.

Secondly, they aim at different problems and cannot be substitutions of each other. For the skill valuation task, feature relevance/XAI methods cannot measure skill value that has explicit physical meanings. Specifically, while they can tell the relevance of a skill for a trained salary prediction model with various measurements (e.g., Shapley value [7], propagation-based metrics [8], and gradient [9]), these measurements lack the physical meaning in the application scenario and cannot be regarded as the value of the skill. In contrast, in our approach, we have given explicit definitions of skill value, which is quantitatively linked with job salary and has the same scale as it. Based on our definition, our approach can learn to 1) estimate the skill value and 2) estimate job salary according to skill value, which feature relevance/XAI methods fail to achieve. Meanwhile, our model cannot be applied to measure the feature relevance of an arbitrary machine learning model, which is the aim of feature relevance/XAI methods.

Instead of proposing a new feature decomposition method, the technical contribution of this paper is to propose a novel cooperative framework to train neural network models for unsupervised learning tasks, by quantitatively linking them with a supervised learning task. Indeed, the unsupervised learning tasks widely exist in many fields, where the model is expected to discover hidden patterns from the unlabeled data. Therefore, we believe our model can benefit the broad research community.

We are sorry for the confusion. In the revised version, we have revised the corresponding descriptions in the paper by adding the following descriptions in the “Introduction” Section to further clarify our technical contribution:

Indeed, SSCN provides a novel cooperative framework to train neural network models for knowledge discovery from unlabeled data, by quantitatively linking them with a supervised learning task.

Meanwhile, we have added a “Technical Contribution” Section to further emphasize the general technical contributions of this work as follows:

Since indirect supervision is common in the real-world, we believe that this work not only provides an intelligent and accurate solution for the skill valuation problem but also can be an inspiration for readers who work on data analysis in other fields of applications. Specifically, in many real-world scenarios, obtaining labeled training data is far from an easy job. It is often the case that we can only obtain indirect supervision from a related task. Learning skill valuation model from job salary data is one of these kinds of problems. In this problem, we have no labeled data of skill value, but we have job salary data as indirect supervision information, with the intuition that high skill value usually leads to high job salary. To this end, we proposed a machine learning-based solution that uses neural network with cooperative structure to model the relationship between job and skills, where the

salary prediction is regarded as a cooperative task for training the skill valuation model. In this way, we obtain an effective skill valuation model under the indirect supervision of job salary data.

As for model evaluation, we fully agree that this paper can be further improved with better evaluation metrics for skill valuation. However, since there is no ground truth data of skill value, we fail to achieve quantitative evaluations on the unsupervised skill valuation task, which we regard as a limitation of this work (as discussed in the “Limitation” Section). Therefore, we have given various empirical evaluations to prove the effectiveness of the proposed model. We are grateful for your suggestions. In our future work, we will try to further improve our skill valuation model and seek ground truth for evaluating it.

Comment 6: *Overall, I have the feeling that the paper, after the rebuttal, is still kind of self-contained and is difficult to assess its impact both to the Labour Market domain (regarding the job value prediction task) and also regarding the problem of feature domination assessment (skill domination task).*

Response: Thank you very much for your comments. Please refer to our above responses for the clarification of the impact and technical contributions of our work.

References

- [1] Kim, Y. Convolutional neural networks for sentence classification. In Proceedings of the 2014 Conference on Empirical Methods in Natural Language Processing (EMNLP), 1746–1751 (2014).
- [2] Yang, Z.et al. Hierarchical attention networks for document classification. In Proceedings of the 2016 conference of the North American chapter of the association for computational linguistics: human language technologies, 1480–1489 (2016).
- [3] Dai, Z.et al. Transformer-XL: Attentive language models beyond a fixed-length context. In Proceedings of the 57th Annual Meeting of the Association for Computational Linguistics, 2978–2988 (2019).
- [4] Yang, Z.et al. Xlnet: Generalized autoregressive pretraining for language understanding. In Advances in neural information processing systems, 5753–5763 (2019).
- [5] Devlin, J., Chang, M.-W., Lee, K. & Toutanova, K. BERT: Pre-training of deep bidirectional transformers for language understanding. In Proceedings of the 2019 Conference of the North American Chapter of the Association for Computational Linguistics: Human Language Technologies, Volume 1 (Long and Short Papers), 4171–4186 (2019).
- [6] Liu, Y.et al. Roberta: A robustly optimized bert pretraining approach. arXiv preprint arXiv:1907.11692(2019).

- [7] Lundberg, S. M. & Lee, S.-I. A unified approach to interpreting model predictions. In *Advances in neural information processing systems*, 4765–4774 (2017).
- [8] Bach, S. et al. On pixel-wise explanations for non-linear classifier decisions by layer-wise relevance propagation. *PloS one*10, e0130140 (2015).
- [9] Simonyan, K., Vedaldi, A. & Zisserman, A. Deep inside convolutional networks: Visualising image classification models and saliency maps. In *2nd International Conference on Learning Representations, ICLR 2014, Banff, AB, Canada, April 14-16, 2014, Workshop Track Proceedings* (2014).

Reviewers' Comments:

Reviewer #1:

Remarks to the Author:

This is the second revision of the above mentioned manuscript. In my last review, I only flagged minor concerns and these minor concerns have been adequately addressed by authors in their latest revision.

Again noting that I am not an expert in the underlying methodology, from my perspective the revisions are satisfactory. I congratulate authors to a nice piece of work and recommend acceptance .

Reviewer #2:

Remarks to the Author:

I think the authors made a great effort to improve the paper following the comments at their best. I'm still convinced that the numerosity of job postings observed limit the significance of the results. I would ask authors to clarify this aspect in the paper as well. That said, from a technical perspective, the paper is now sound and clear.

Response to Reviewers

Title: Market-oriented Job Skill Valuation with Cooperative Composition
Neural Network

Submission ID: NCOMMS-20-27613d

Authors: Ying Sun, Fuzhen Zhuang, Hengshu Zhu, Qi Zhang, Qing He, and
Hui Xiong

First of all, we would like to thank the editor and two anonymous reviewers again for providing us an opportunity to further revise this paper. Their valuable comments and constructive suggestions have not only helped a lot on the improvement of this paper in a number of ways, but also given a lot of inspirations for our future research works. In the following, we address the specific comments from the two reviewers. As a convention, the “last submitted version” refers to the version we submitted in January 2021 for the third-round review, and the “revised version” refers to the current submitted version.

1. Response to Reviewer 1

Comment 1: *This is the second revision of the above mentioned manuscript. In my last review, I only flagged minor concerns and these minor concerns have been adequately addressed by authors in their latest revision. Again noting that I am not an expert in the underlying methodology, from my perspective the revisions are satisfactory. I congratulate authors to a nice piece of work and recommend acceptance .*

Response: Thank you very much for your recognition on this paper and all your valuable comments on improving this paper!

2. Response to Reviewer 2

Comment 1: *I think the authors made a great effort to improve the paper following the comments at their best.*

Response: Many thanks for your valuable comments and your recognition on this paper!

Comment 2: *I'm still convinced that the numerosity of job postings observed limit the significance of the results. I would ask authors to clarify this aspect in the paper as well.*

Response: Thank you for your suggestion. We have modified the Limitations section to clarify the limitation of this work on data size as follows:

On the other hand, since our research has certain requirements on the data quality (e.g., detailed skill requirement, job salary and contextual information), in this paper we only evaluated our model with two datasets collected from one of the largest and most popular Chinese online recruitment website of Internet-related industry. This may bring bias to the analysis. If provided with more large-scale and comprehensive data, our model will obtain more significant insights.

Comment 3: *That said, from a technical perspective, the paper is now sound and clear.*

Response: Many thanks for your recognition on the technical soundness of this paper!